# A prudent planetary limit for geologic carbon storage

Matthew J. Gidden[1,2 ✉], Siddharth Joshi[1], John J. Armitage[3], Alina-Berenice Christ[3], Miranda Boettcher[4,5], Elina Brutschin[1], Alexandre C. Köberle[6,7,8], Keywan Riahi[1], Hans Joachim Schellnhuber[1], Carl-Friedrich Schleussner[1,9] & Joeri Rogelj[1,10]

Geologically storing carbon is a key strategy for abating emissions from fossil fuels and durably removing carbon dioxide ($CO_2$) from the atmosphere[1,2]. However, the storage potential is not unlimited[3,4]. Here we establish a prudent planetary limit of around 1,460 (1,290–2,710) Gt of $CO_2$ storage through a risk-based, spatially explicit analysis of carbon storage in sedimentary basins. We show that only stringent near-term gross emissions reductions can lower the risk of breaching this limit before the year 2200. Fully using geologic storage for carbon removal caps the possible global temperature reduction to 0.7 °C (0.35–1.2 °C, including storage estimate and climate response uncertainty). The countries most robust to our risk assessment are current large-scale extractors of fossil resources. Treating carbon storage as a limited intergenerational resource has deep implications for national mitigation strategies and policy and requires making explicit decisions on priorities for storage use.

Storing carbon for centuries to millennia in geologic formations will be required if the world is to achieve the goals of the United Nations (UN) Paris Agreement. Key milestones include reaching net-zero carbon dioxide ($CO_2$) emissions, at which point global temperatures are likely to stop increasing[5,6], and pursuing mitigation strategies to reach net-negative $CO_2$ and net-zero greenhouse gas (GHG) emissions, at which point global mean surface temperature could start to decline[7]. This need has been recognized by scientists and policymakers for decades[8], including in recent UN climate negotiations[9].

Net-zero $CO_2$ emissions will occur when gross sources of $CO_2$ equal removals by sinks. Carbon capture and storage (CCS) has a role in both reducing sources (through, for example, storing captured $CO_2$ from cement production and fossil-fuel combustion) and durably removing $CO_2$ from the atmosphere (for example, storing $CO_2$ captured from the atmosphere or biomass combustion). Despite its prominence in scientific and policy discourse, present deployment of CCS is small, with 49 $MtCO_2$ $yr^{-1}$ of capture capacity in operation and 416 $MtCO_2$ $yr^{-1}$ either planned or in construction[10]. Almost all facilities are planned to store carbon in sedimentary rock formations, the focus of our analysis, with the largest planned basalt-based storage facility having a capacity of less than 0.036 $MtCO_2$ $yr^{-1}$. The majority of operational storage facilities are utilized for enhanced oil recovery, a process that overall results in net-positive $CO_2$ emissions[11]. However, the technical potential for geologic carbon storage is commonly assumed to be vast, with estimates of available storage of around 10,000–40,000 $GtCO_2$ in the scientific literature[3,12]. Industry estimates are around 14,000 $GtCO_2$, of which 13,400 $GtCO_2$ is undiscovered[4] and just 253 $MtCO_2$ is considered currently economically viable[13].

Many scenarios that limit climate change to the goals set out by governments in the Paris Agreement[1] assume a large scale-up of the use of CCS to abate further combustion of fossil fuels, reduce emissions from industrial sectors that have limited or no mitigation alternatives, and durably store $CO_2$ that was removed from the atmosphere, contributing to carbon dioxide removal (CDR). Planning to use storage for both reducing sources of emissions and for CDR presents risks should storage infrastructure fail to be deployed at scale. This risk can be somewhat mitigated by sustainably deploying a portfolio of approaches, including storing carbon in the built environment through what is referred to as the forestry–construction pump[14], enhancing the carbon content of soils, and conserving and expanding land and coastal carbon sinks, which would reduce dependence on geologic storage by using less-durable carbon-storage media.

The scale of deployment of CCS (and thus geologic storage) in future scenarios is not absolute and depends on policy and political choices. While nearly all scenarios limiting warming below 2 °C deploy some level of CCS, some scenarios that exceed the 1.5 °C warming limit by a large margin and subsequently hope to reverse global warming utilize up to 2,000 $GtCO_2$ of storage by the end of the century. The higher the so-called overshoot of a specific acceptable level of warming, the bigger the need for atmospheric $CO_2$ removal and hence cumulative carbon storage. Demand for geologic carbon storage can also increase in these scenarios based on other mitigation choices, such as deploying blue (fossil-gas based) versus green (renewable-energy based) hydrogen, electrifying steel production versus capturing carbon from existing processes, and reducing demand for cement versus capturing carbon from cement production. Even more storage will be needed than shown

[1]International Institute for Applied Systems Analysis, Laxenburg, Austria. [2]Center for Global Sustainability, University of Maryland, College Park, MD, USA. [3]IFP Energies nouvelles, Earth Sciences and Environmental Technologies Division, Rueil-Malmaison, France. [4]German Institute for International and Security Affairs (SWP), Berlin, Germany. [5]Copernicus Institute of Sustainable Development, Utrecht University, Utrecht, The Netherlands. [6]Instituto Dom Luiz (IDL), Faculdade de Ciências, Universidade de Lisboa, Lisbon, Portugal. [7]Potsdam Institute for Climate Impact Research, Potsdam, Germany. [8]Centre for Climate Finance and Investment, Imperial College Business School, London, UK. [9]Geography Department and IRITHESys Institute, Humboldt-Universität zu Berlin, Berlin, Germany. [10]Centre for Environmental Policy and Grantham Institute – Climate Change and Environment, Imperial College London, London, UK. ✉e-mail: gidden@umd.edu

in scenarios if the Earth system responds differently than expected by the current state-of-the-art climate models[15].

Large-scale utilization of carbon storage comes with sizable risks and deep uncertainty of feasible storage potential and injection rates[16,17], which are not well captured in models that describe how future emissions reductions could be achieved. Leakage of $CO_2$ from storage sites due to seismic activity, well-head failure or other factors would potentially reintroduce carbon into the atmosphere[18], where annual leakage rates greater than 0.01% can negate the climate benefits of stored $CO_2$ (refs. 18,19). For storage locations near population centres, the potential seepage of stored carbon into municipal aquifers can change groundwater quality and pose health concerns[20]. These risks and other considerations such as environmental conservation, risk of project failure[21], public perception and the geopolitics of transnational sedimentary basin boundaries[22] could severely restrict the total available carbon-storage volume that can be assumed when developing national energy plans and climate strategies.

The Paris Agreement establishes a number of specific requirements for parties when making climate pledges, including that they are fair, ambitious and in line with the best available science. Norms of international law go further, requiring "high standards of due diligence to prevent transboundary environmental harm"[23]. Although previous studies have estimated global[3,24] or regional[25,26] technical storage potential, so far, no consistent estimate of the global carbon-storage potential assesses this variety of different risk factors to determine the available storage potential from a precautionary harm-prevention perspective as expected under the UN Framework Convention on Climate Change. Country plans for utilizing carbon storage in their climate strategies can thus at present not be evaluated against potential risks and storage limits. When assessed globally, our estimates can inform an understanding of prudent and precautionary planetary limits for geologic carbon storage.

Here we provide a spatially explicit estimation of carbon-storage potential in sedimentary basins consistent with the principle of harm prevention, which can guide policymakers in considering the amounts of $CO_2$ storage in their emissions targets that are robust to multiple sources of uncertainty and risk. We argue that nations should make explicit plans for geologic carbon storage for both mitigating continued sources of fossil-fuel emissions as well as durably removing and storing $CO_2$. Treating geologic carbon storage as a limited global resource that should be managed on an intergenerational timescale requires considering the trade-offs of continuing to emit carbon from fossil-based sources versus utilizing storage space for removing carbon from the atmosphere to ultimately lower the global mean temperature for this and future generations.

## Limits to geologic carbon storage

We conceptualize a planetary limit for geologic carbon storage in the context of harm prevention and risk avoidance. When carbon storage is understood as a consumable and depletable common good, transgressing this limit results in permanent trade-offs with other dimensions of sustainable development such as human health and biodiversity[27]. Future impacts of climate change, their potential reversibility and respective consequences will be critically dependent on this limit.

We perform a spatially explicit analysis, identifying and systematically applying multiple exclusion layers based on preventative risk analysis to a global map of sedimentary basins suitable for carbon storage (Methods, Supplementary Figs. 1–6 and Supplementary Tables 1–3, including for assessed sensitivities). We then estimate a planetary limit for geologic carbon storage by assessing the remaining onshore and offshore basin volumes that meet all risk criteria (Fig. 1).

## Geologic storage and geophysical risk

The most prevalent present-day target sites for $CO_2$ storage are depleted hydrocarbon fields or deep saline aquifers within geologically stable sedimentary basins. Several criteria must be considered when assessing the suitability of basins for $CO_2$ storage, such as storage capacity based on pore volume, depth of target formation, seal integrity, tectonic hazards and basin type[28,29]. A typical high-quality reservoir for $CO_2$ storage would have presence of a seal (layers of impermeable cap rock), favourable petrophysical parameters for injectivity and storage, sufficient depth, and low risk for reactivation of existing faults. Cautionary approaches restrict injection to minimum depths of about 1 km to ensure that the $CO_2$ is in a supercritical state and maximum depths of about 2.5 km to avoid destabilizing bedrock and to limit potential seismic activation of deep-rooted faults[30], which we use to constrain our volumetric calculation. We also limit our central estimate to ocean depths of 300 m or less, where the vast majority of current offshore oil and gas infrastructure are predominantly sited, owing to both economic and risk considerations. We assess uncertainty ranges for storage depth based on an extensive literature review and ocean depth based on a geographic information system analysis of current oil and gas infrastructure (Methods).

Basins in the proximity of active seismic zones, for example, close to plate subduction areas, have elevated in situ stress, making them prone to complex fault systems and tectonic events. The pressure increase during $CO_2$ injection can lead to induced seismicity via fault reactivation, potentially triggering low-intensity earthquakes. Furthermore, fault reactivation can also compromise the storage complex, creating pathways for the $CO_2$ to escape[31]. As such, we exclude basin areas where historic seismic activity is larger than 'moderate' severity based on US Geological Society's scale[32].

## Environmental and human risk

Following a long history of environmental protection and conservation, and in line with international agreements such as the Kunming–Montreal Global Biodiversity Framework, we exclude sensitive environmental areas based on protected areas as well as the Arctic and Antarctic polar circles.

We further exclude a 25-km buffer area (central case) around built-up areas of human settlement under a high-population future scenario (Methods) to minimize the risks for human health from direct leakage from aboveground infrastructure or release of carbon from the reservoir in which it will be stored for centuries to millennia. $CO_2$ that escapes to the surface can pose a threat to shallow groundwater reservoirs by lowering the pH of the groundwater through the formation of carbonic acid. This might have several secondary effects, for example, the mobilization of toxic metals, sulfate or chloride[33], which may contain impurities of other gases, such as hydrogen sulfide, sulfur dioxide or nitrogen dioxide, which increase the effect of toxic metal mobilization[34].

## Policy risk

These environmental considerations and perceived risks, as well as general concerns about delaying the scale-up of renewables and the perception that CCS may prolong the use of fossil fuels, have been linked to low levels of public and political support of geologic carbon storage[35,36]. For example, CCS is currently banned or majorly restricted in some European countries (see Supplementary Table 4 for current countries with restrictive CCS policies), but there are growing discussions to adjust the existing regulations to allow onshore and offshore storage to achieve climate targets. However, these policy developments remain politically contested, highlighting the volatile and uncertain nature of public and political support of geologic carbon storage.

If supported by domestic policy, countries have legal authority to store carbon within their own boundaries (including, for example,

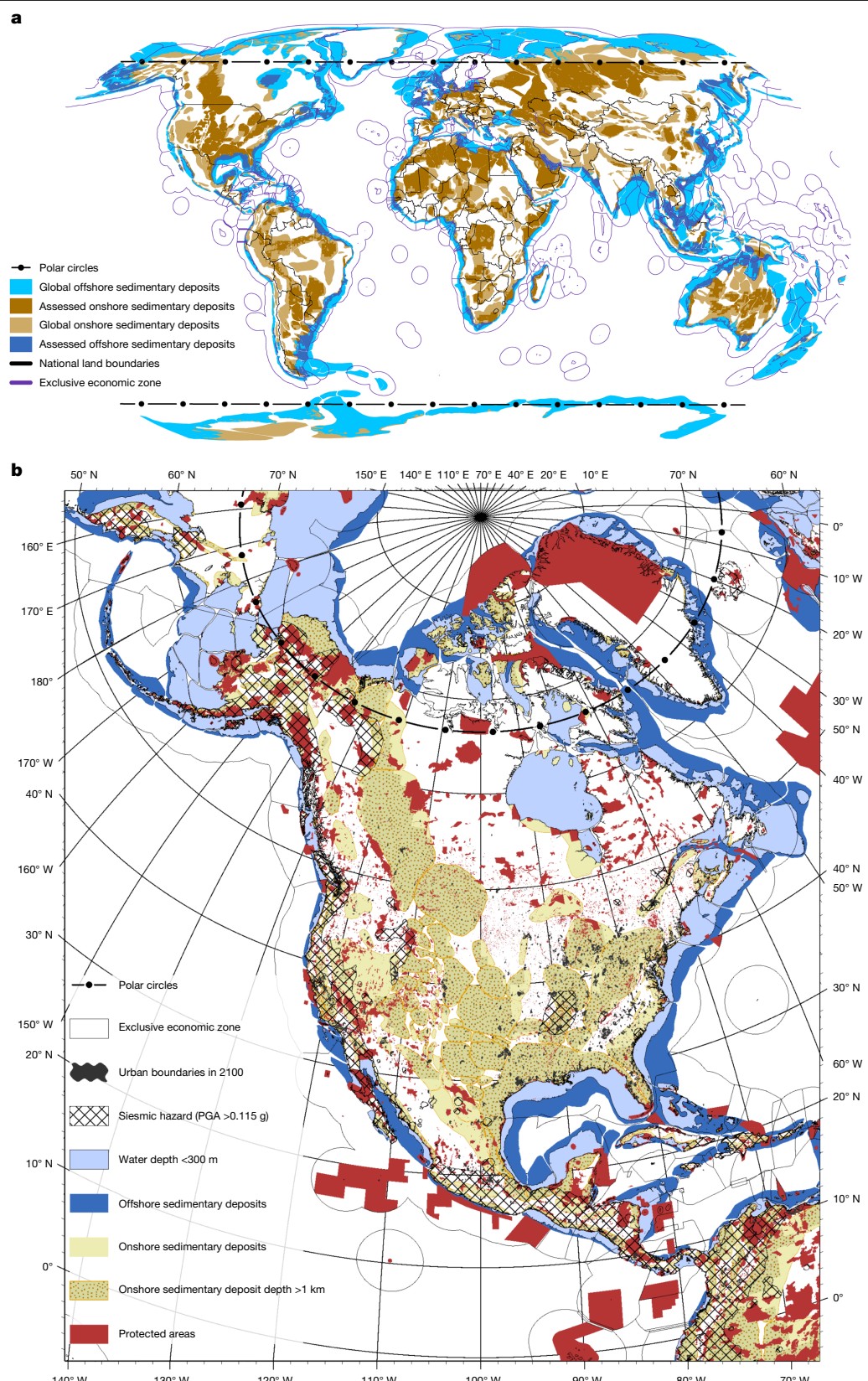

**Fig. 1 | Spatially explicit global carbon-storage potential in sedimentary basins. a**, Onshore (brown) and offshore (blue) sedimentary basins, including national terrestrial and maritime borders (that is, EEZs). Basin colours vary according to technical carbon-storage potential (lighter) and the assessed prudent carbon-storage potential (darker). **b**, The North American continent, including all exclusion layers (Supplementary Table 1). The prudent limit is estimated by accounting for the full storage technical potential, removing all precautionary exclusion layers and summing up available carbon storage from the basins that remain (yellow dotted and light blue areas). PGA, peak ground acceleration. **a**,**b**, Sources: Esri, GEBCO, NOAA, National Geographic, DeLorme, HERE, Geonames.org and other contributors.

their marine exclusive economic zones (EEZs)). However, it remains largely unclear how international treaties would perform in a future with multiple state actors injecting carbon into a common basin, on land but especially at sea, given a recent advisory opinion by the International Tribunal for the Law of the Sea[37]. The London Convention and London Protocol has, so far, been the most proactive international body addressing carbon storage in subsea geologic formations[38], having adopted a resolution permitting countries to bilaterally agree on sharing such transboundary storage. So far, though, only very few countries have applied for such an allowance[39]. We include all basins within EEZs exclusive of contested territorial claims in our central assessment with certain restrictions in our sensitivity case, but note that any policy outcome that severely limits offshore storage locations would strongly limit the total global storage potential.

### A holistic risk assessment

We find that the initial global physical storage potential of 11,800 GtCO$_2$ is reduced by about an order of magnitude after combining all our spatial risk layers to a planetary limit of 1,460 GtCO$_2$ (1,290–2,710 GtCO$_2$), of which about 70% occurs onshore and about 30% occurs offshore (Fig. 2a–c and Supplementary Table 1). Our multi-dimensional risk-prevention approach results in heterogeneous outcomes across countries (Fig. 2d–f and Supplementary Table 5). Many countries with rich natural carbon-storage reserves maintain relatively high levels of storage potential even after our risk analysis, notably, Russia, USA, China, Brazil and Australia. A number of countries have high levels of storage potential that are left largely untouched by our risk analysis, including Saudi Arabia, the Democratic Republic of the Congo and Kazakhstan. Other regions see large decreases in available storage affecting a significant portion of their total storage potential, with India, Norway, Canada and countries within the European Union experiencing the largest impact. When we apply exclusion layers in the order presented here, we find that that the largest increase in storage would be realized if our assumptions regarding storage and ocean depth were relaxed, followed by assumptions regarding storage in polar regions and protected areas (Supplementary Table 1).

## Implications for future mitigation strategies

The majority of mitigation strategies consider geologic carbon storage to some extent in support of the transformation towards net-zero and net-negative CO$_2$ futures[2]. Which carbon capture approaches are utilized in future mitigation scenarios depends on a variety of factors, including assumed costs, scale-up rates and the efficiency of capture. Coupling carbon capture with hydrogen and synthetic fuel production provides efficient pathways to achieve deep mitigation in heavy industry and transportation sectors[40] and can enable net-negative sectoral outcomes when using biomass instead of fossil feedstocks[41]. Future scenarios tend to utilize large-scale carbon capture at individual point sources in the power (for example, biomass and fossil-fuelled generation) and industry (for example, cement production) sectors owing to the relatively high concentrations of carbon in the effluent flue gases[42]. Net-negative emissions futures are increasingly being studied that utilize direct air capture with CCS, which removes and durably stores ambient CO$_2$ from the atmosphere[43–46]. The regional allocation of the ultimate storage depends on assumptions of regional storage capacity and infrastructure needs, which vary in their level of detail across different modelling frameworks[47].

Cumulative storage demand in scenarios is driven by factors including the peak temperature achieved in a scenario, the ultimate level of temperature decline thereafter, and the chosen mitigation strategy regarding the phase out of the use of fossil fuels, be it abated or unabated. The amount of geologic storage needed at the point of net-zero CO$_2$ emissions is a function of the amount of fossil-energy emissions abated through CCS, the remaining positive (residual) CO$_2$ emissions,

and the amount of carbon removal owing to conventional, land-based methods such as reforestation and soil carbon sequestration. Scenarios tend to use storage primarily for carbon removal and fossil point-source capture, with industrial capture having an important but smaller role (Supplementary Fig. 7). The durability of carbon sequestered via geologic storage tends to be significantly better compared with land-based removals as geologically stored carbon is not subject to the same environmental and human factors that can lead to leakage of land-based carbon removals into the atmosphere[48], a distinction that is not generally considered in mitigation scenarios or strategies.

To maintain net-zero and achieve net-negative CO$_2$ emissions implies a persistent demand for storage resources (Fig. 3a) based on the continued use of abated fossil fuels and residual emissions for five primary purposes: counterbalancing residual emissions from fossil-fuel use (limiting CO$_2$ pollution and temperature increase); counterbalancing other residual CO$_2$ (limiting CO$_2$ pollution and temperature increase); counterbalancing residual long-lived non-CO$_2$ GHGs (limiting climate forcing and temperature increase); achieving net-negative CO$_2$ emissions beyond point 2 (reversing CO$_2$ pollution and enhancing temperature decrease); and achieving net-zero total GHG emissions (a key milestone in the Paris Agreement, reversing CO$_2$ pollution and enhancing temperature decrease).

Nearly all 2-°C-and-lower temperature scenarios assessed by the Intergovernmental Panel on Climate Change (IPCC) stay within at least a 50% margin of our assessed planetary limit when net-zero CO$_2$ emissions are reached (Fig. 3b). Even so, scenarios limiting warming to 1.5 °C with no or limited overshoot (>33% to stay below 1.5 °C until 2100, >50% in 2100) sequester 8.7 (5.9–13) GtCO$_2$ yr$^{-1}$ when reaching net-zero CO$_2$ emissions around 2050–2055. This represents a 175-fold increase from today's levels and an industrial capacity on par with current global crude oil production[49]. Carbon injection rates at net-zero CO$_2$ systematically increase with decreasing policy stringency as delayed near-term mitigation action results in stronger dependence on carbon storage.

Maintaining this level of storage (as in the limited fossil, stabilization scenario of Fig. 3a) results in an eventual breach of our proposed threshold in the next 250 years across more than 75% of all assessed scenarios (Fig. 3c). Scenarios with the strongest climate action take the longest time to reach this limit, on average about 150 years after reaching net-zero CO$_2$ emissions. Scenarios of less stringent climate action reach this limit earlier, with scenarios limiting warming to likely 2 °C (that is, with >67% probability), reaching it on average about 120 years after net-zero CO$_2$.

Much more storage will probably be needed after net-zero CO$_2$ emissions is achieved to help draw down global mean temperatures by continuing to counterbalance residual emissions and actively remove CO$_2$ from the atmosphere. Although there is no agreed-upon limit to global temperature stabilization, the Paris Agreement and subsequent UN decisions outline that 1.5 °C is a threshold that should be returned to—if breached. A proportion of scenarios assessed by the IPCC that achieve net-zero CO$_2$ emissions this century breach our proposed planetary limit by the end of the century (Fig. 3b) to enable temperature drawdown. In 2100, carbon storage activity is continuing to grow, with 1.5-°C and 2-°C scenarios storing on average 15 (11–18) GtCO$_2$ yr$^{-1}$. The exceedance is not regionally uniform, with more than 50% of all scenarios breaching our assessed limit in the IPCC's Asia region (Supplementary Fig. 8, and Supplementary Figs. 9–12 for other regions), which includes some of the largest emerging economies with high levels of current emissions and future emissions under current climate policies and targets[50,51], such as China and India.

Crucially, although scenarios describe mitigation pathways until the end of the century, they represent a continued future beyond this time horizon under which limitations on carbon storage persist[52]. The continued demand for storage resources can proceed indefinitely if either fossil resources continue to be consumed or there is a need to deploy CDR to further draw down temperature. Following the

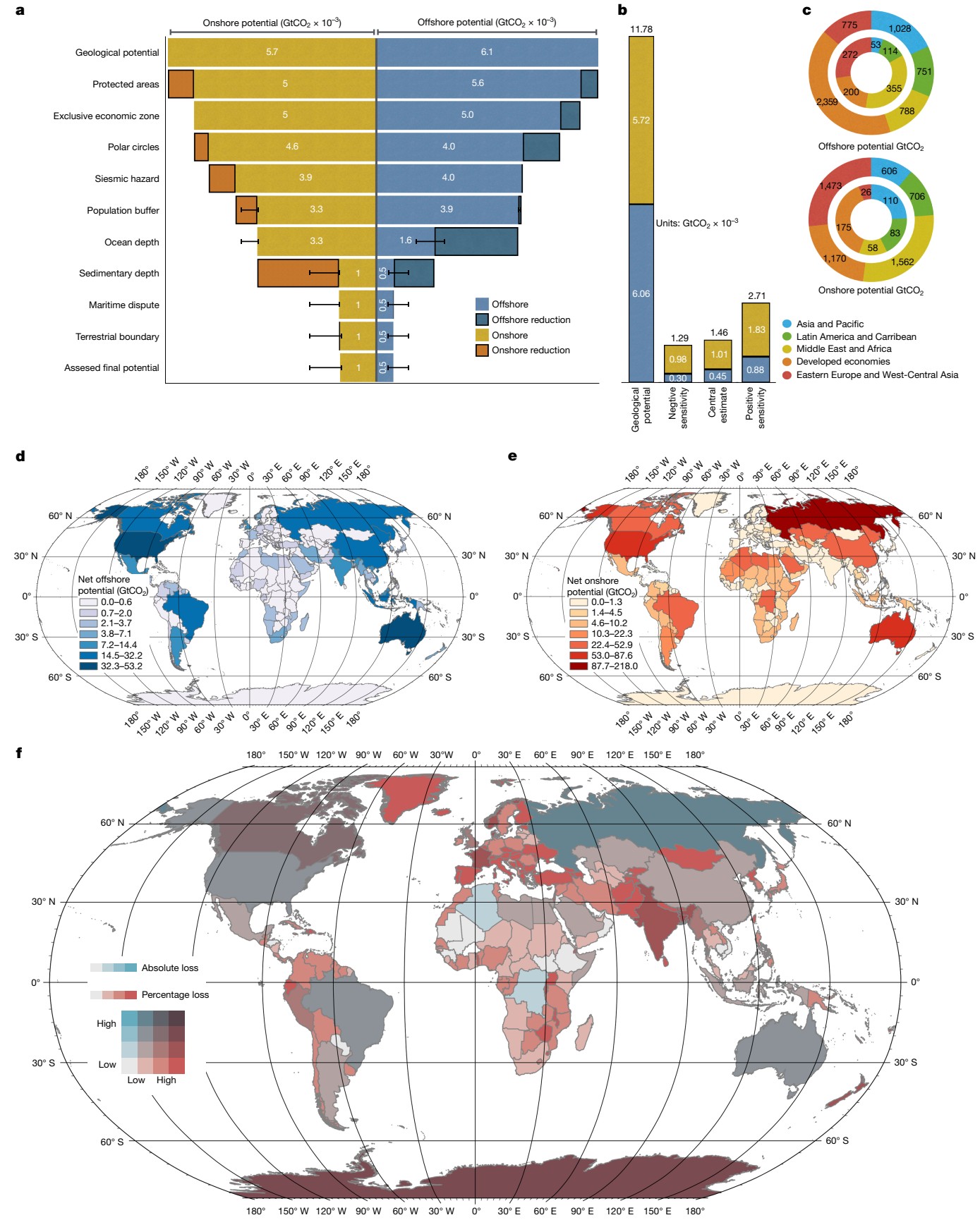

**Fig. 2 |** See next page for caption.

**Fig. 2 | Storage potential loss due to application of different risk layers.**
**a**, The reduction in global storage potential after applying each subsequent exclusion layer (Supplementary Table 1). Assessed sensitivities form the lower and upper values of each uncertainty bar around the central estimate (Supplementary Table 2). **b**, The difference in total global potential before (left) and after (right) all exclusion layers are applied, resulting in the assessed planetary limit, including the central estimate and sensitivity cases. **c**, The full technical potential and final assessed potential in the central estimate by IPCC region (Supplementary Table 7). **d**,**e**, Our analysis is spatially explicit and globally consistent, allowing for country-level assessments of prudent storage potential in both offshore (**d**) and onshore (**e**) basins. **f**, Total storage before and after applying precautionary exclusion layers is heterogeneous across countries based on total storage magnitude loss (light colours to dark colours) and the percentage of technical potential lost (blue represents high absolute loss but low percentage loss, red represents high percentage loss but low absolute loss, and purple represents high loss along both axes). **d**–**f**, Sources: Esri, GEBCO, NOAA, National Geographic, DeLorme, HERE, Geonames.org and other contributors.

end-of-century carbon-storage trends in scenarios shows that nearly all scenarios would exceed available storage in basins with existing oil and gas infrastructure by 2125 and would exceed the planetary limit for geologic storage before 2200 (Fig. 3d).

Our prudent limit to geologic carbon storage of 1,460 $GtCO_2$ brings profound implications for robust mitigation strategies that depend on the most durable forms of carbon storage. This limit sets a cap of about 0.7 °C (0.6–1.2 °C based on assessed exclusion criteria uncertainties; Supplementary Table 2) on the total warming that can ever be sustainably reversed if the full prudent potential is used for durable $CO_2$ removal, assuming the IPCC's central estimate of the temperature response to cumulative emissions of $CO_2$ (0.45 °C per 1,000 $GtCO_2$) and that climate response to net-negative emissions is similar in magnitude to its response to net-positive emissions, which might not always be the case[53]. Any share of this prudent potential that is used for continued fossil-fuel use with CCS reduces this maximum amount proportionally, as does any level of residual $CO_2$ and other long-lived non-$CO_2$ emissions that are not eliminated.

Taking a more precautionary interpretation of the climate response to cumulative emissions of $CO_2$ and its effectiveness in reversing warming shows an even starker outcome. Assuming the lower end of the assessed likely range of temperature response (that is, the lower end of the central 66% range) of 0.27 °C per 1,000 $GtCO_2$ indicates that the prudent potential of geologic carbon storage at best can facilitate a temperature reversal from peak warming of about 0.4 °C (0.35–0.7 °C), which is by all means an overestimate of the real-world reversal potential given the anticipated persistence of residual emissions from several industrial and agricultural activities.

## Robust strategies under uncertainty

Recognizing that geologic carbon storage may be a limited resource requires careful consideration to be taken by nation states when developing domestic energy transition and climate plans. Given the millennial timescales for which carbon storage is needed to counteract the impact of released $CO_2$ on climate change, decisions made on carbon management today will affect the human population for more than ten generations into the future. This raises the question of which countries, sectors and generations should be entitled to utilize available geologic storage resources.

There is unequal impact on countries' geologic carbon-storage reserves across dimensions of both capabilities to mitigate and historic responsibility for emissions based on our analysis (Fig. 4a). Some high-gross-domestic-product countries with high historic emissions such as Russia, USA and Canada are better placed to implement geologic storage solutions, whereas other relatively rich historic emitters such as the European region have substantially reduced storage potential. At the same time, some developing countries with robust storage potential such as Indonesia and Brazil, and some countries in Africa, have historically contributed little to global emissions and thus may have weak domestic incentive to exploit their storage resources unless the removals can be traded. In a future that significantly exploits available storage resources, our results indicate that large-scale transfers of captured carbon may occur, resulting in higher risks for leakage during transmission, either through shipping or via pipelines, and raising question of distributive justice and equity.

We also find that oil-producing nations in the Arabian Peninsula, who have the know-how to pursue carbon storage also have storage reserves that are largely robust to our risk-prevention analysis. Other countries with a long history of an active domestic oil and gas industry and also relatively large storage potential include USA, Australia and Canada. For global policy limiting warming to well below 2 °C to be successful, these incumbent industry actors must be appropriately incentivized to become net injectors, rather than extractors, of subsurface carbon. Pre-existing norms in international climate agreements, such as the polluter-pays principle, provide avenues for establishing needed financial frameworks.

Even with well-designed policies and markets that incentivize such a reversal in business models, large-scale industries situated in storage-rich nations can develop financial flows in the billions to trillions of dollars per year, which could enhance inequality between and within nations[54]. Still, opportunities exist to enable growth in removing carbon from the atmosphere and storing it today based on principles of fairness, responsibility and respective capabilities. For example, countries with large sovereign funds based on oil and gas revenues could set aside a small portion of their valuations to support the nascent CDR industry[55].

Although we focus on carbon storage in sedimentary basins because of their desirable storage properties[56] and long experience of their exploration by the oil and gas sector[57], our estimates of a prudent planetary limit would expand beyond our explored sensitivities (Supplementary Tables 1 and 2) if other carbon sequestration media become available. Perhaps most promising is the sequestration of carbon through mineralization in basalt formations, as is being piloted at the CarbFix injection site in Iceland[58] and the USA[59]. The potential size of this storage media is highly uncertain with strong dependencies on site-specific characteristics[60]. At present, this technology is still in the development phase, having stored since its operations in 2014 a total of $10^{-4}$ $GtCO_2$.

However, our assessment also does not consider the potentially large barriers to scaling up a carbon-storage industry to the gigaton scale as depicted by most future scenarios (Supplementary Fig. 13). For example, we do not take into account explicitly the substantial governance challenges that are faced by large-scale deployment of carbon storage, including incentive structures and trade-offs with other Sustainable Development Goals[61] nor the distributive justice[62] and equity implications of storage locations[63]. We also do not consider long timescales and high costs for subsoil characterization and seismic surveying required for identifying areas of highest potential injectivity rates[16]. Although we focus on volumetric limits, a large body of recent literature also highlights concerns regarding the technoeconomic feasibility of scaling up subsurface injection to the levels shown in future scenarios[16,21,47], with most scenarios breaching assessed feasibility limits, although some argue that these levels are geophysically feasible[64]. More explicit consideration of injection feasibility would probably further reduce our estimate of usable storage potential.

Most critically, though, the scenarios we assess do not account for the substantial uncertainties in the climate system response to

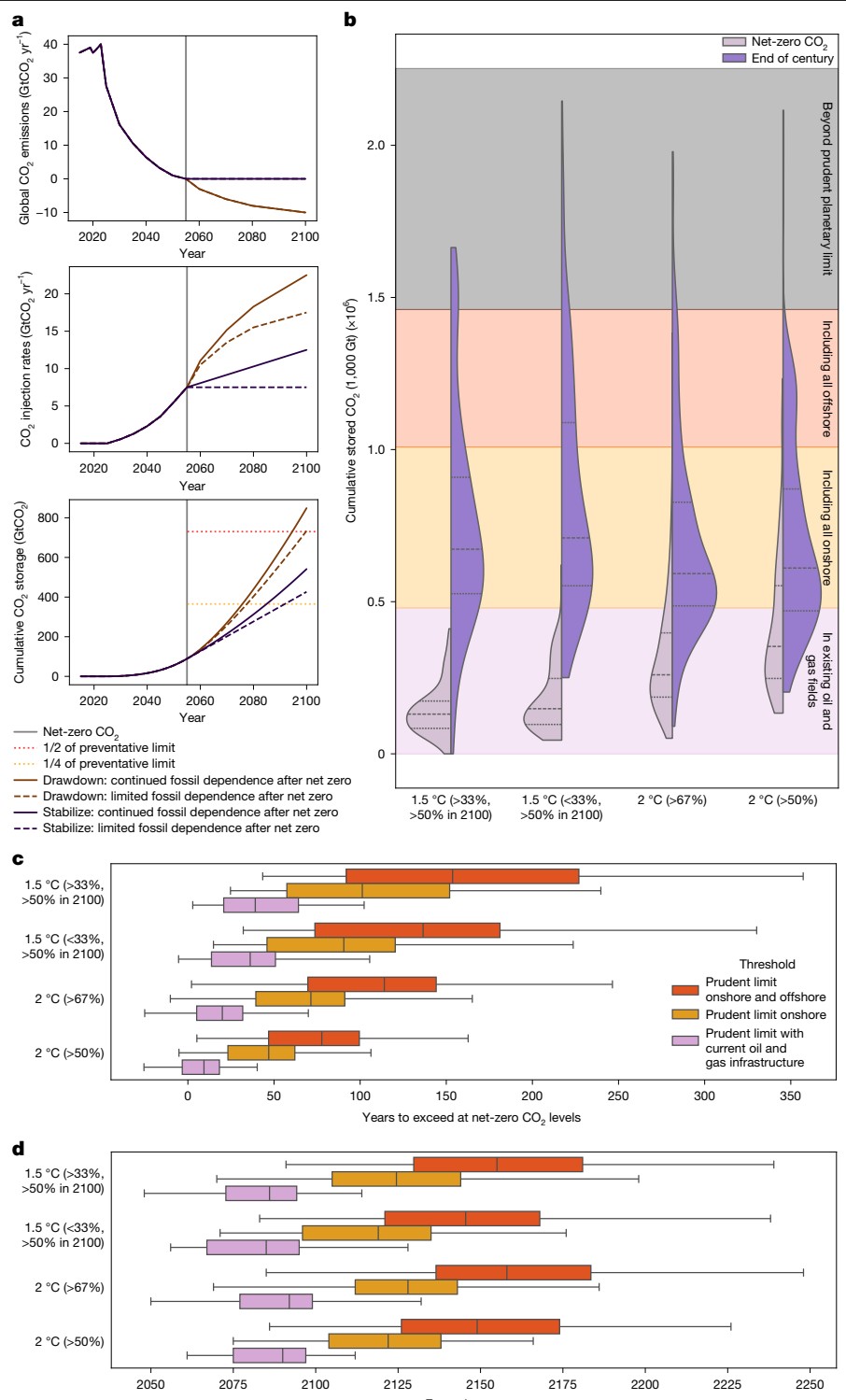

**Fig. 3 | Geologic carbon storage in scenarios exceed the prudent planetary limit. a**, Schematic highlighting the assumed use of carbon storage in mitigation strategies based on a future trajectory of net $CO_2$ emissions and whether a temperature limit is achieved and maintained or whether a limit is exceeded after a peak and temperature drawdown occurs thereafter (top). Total yearly carbon storage is further differentiated by the strategy of fossil-fuel consumption pursued towards and after achieving net-zero $CO_2$ emissions (middle and bottom). **b**, Cumulative stored carbon (scale of 1,000 $GtCO_2$) distributed across scenarios until the time of net-zero $CO_2$ emissions (left-side distributions) and until the last modelled year (2100, right-side distributions) against different thresholds: all non-excluded basins that currently have operational oil and gas facilities (purple), additional storage consistent with all

remaining onshore basins (yellow), remaining offshore basins (red) and the prudent planetary limit (grey). In each distribution, the full range, median and interquartile lines are shown. **c**,**d**, The number of years it would take to reach each of the shown limits if storage levels were maintained after achieving net-zero $CO_2$ emissions (**c**) or if storage levels were extrapolated beyond the year 2100 (**d**). The bars represent interquartile ranges and whiskers represent the 5th–95th percentile in **c** and **d**. Although we aggregate thresholds globally here, carbon storage is regionally deployed in integrated models with different regions exceeding thresholds at different points in time (Supplementary Figs. 8–12), with storage in the IPCC's Asia region (including China and India) exceeding even our planetary limit threshold this century (Supplementary Fig. 8).

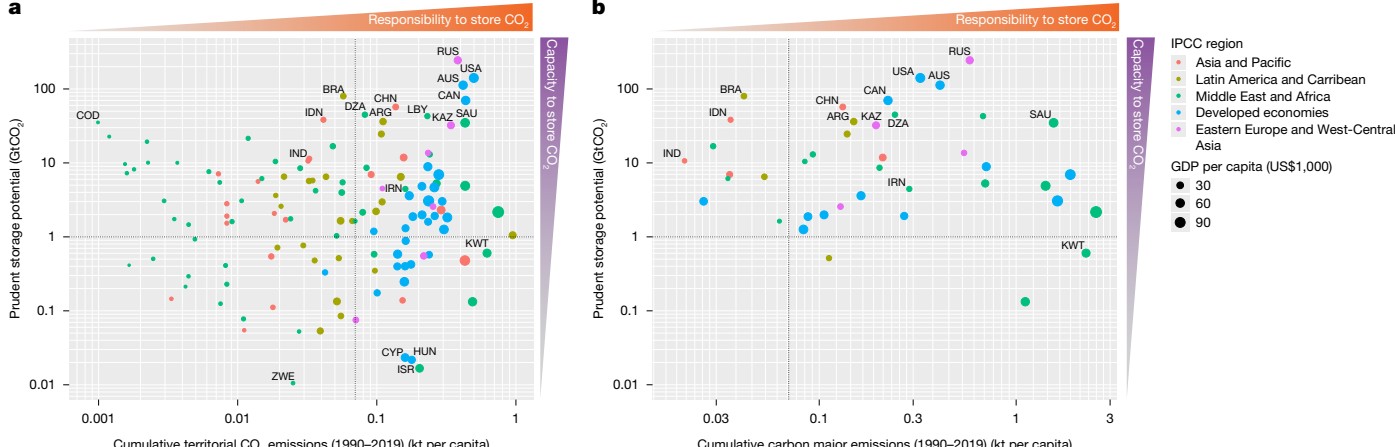

**Fig. 4 | Prudent carbon-storage potential is unequally distributed among countries. a**, The relationship between responsibility for historical emissions (*x* axis)[66] and the remaining storage potential (*y* axis) is shown, with the size of each point in the scatter plot being proportional to per capita gross domestic product (GDP). Countries in the top-right quadrant of the plot have relatively high historical responsibility for current warming levels and relatively high amounts of carbon storage available to support high-durability carbon removal, which in principle would lead to reducing future responsibility. Countries in the bottom-right quadrant have high responsibility, but limited capacity to store carbon domestically based on our analysis, implying the need to transport carbon elsewhere. Countries in the top-left quadrant have low

responsibility but high remaining storage potential, and thus could in principle provide storage for appropriate financial transfers to countries without available resources under the Paris Agreement's principle of common-but-differentiated responsibilities and respective capabilities to arrive at fairer outcomes aligned with its long-term temperature goal. Individual three-letter country codes are provided in Supplementary Table 5. **b**, The same plot as in **a** but using the emissions embedded in extracted fossil fuels by industrial carbon majors (https://carbonmajors.org/Downloads), showing which nations have the highest responsibility for historical fossil-fuel extraction—and thus who has reaped the largest revenues from sale of fossil resources—rather than territorial emissions.

continued GHG emissions, which may require several hundred gigatons of additional carbon-storage capacity to meet expected temperature outcomes[15]. There is still uncertainty about whether removing one unit of net carbon reverses warming to the same extent that emitting one unit of net $CO_2$ increases it[53]. This potential asymmetry is further compounded by the possibility of several tenths of a degree of additional warming occurring after global $CO_2$ emissions reach net-zero levels[6]. Should this uncertainty in the climate response work to our disadvantage, substantially more carbon than currently estimated will need to be removed to reach desired climate outcomes[15,65]. Each of these considerations would further limit the amount of carbon storage that should be used by policymakers in planning their long-term climate strategies.

Our findings highlight the critical importance of transparency in carbon management planning and motivates treating geologic carbon storage as a scarce resource that needs to be deployed strategically to maximize climate benefits rather than treating geologic carbon storage as a limitless commodity. For example, understanding whether nations plan to maximize their use of storage resources for abating continued sources of emissions that could be avoided (for example, through the pursuit of blue, fossil-based hydrogen and fossil CCS) or strategically minimize the dependence of their climate strategies on carbon storage (for example, by deploying green, renewable-based hydrogen, other renewable-energy strategies and minimal CDR) will enhance understanding of the robustness of mitigation plans. Policymakers can make explicit their expectations for utilizing geologic carbon storage in their national energy transition plans, nationally determined contributions and long-term strategies, and communicate the degree to which these plans address environmental and societal risks when allocating what is a fundamentally finite resource.

Applying our prudent planetary limit framework demonstrates that following current climate policies will not only overshoot the 1.5-°C limit of the Paris Agreement by a wide margin but also may prohibit returning to it thereafter. Robust mitigation strategies are needed that weigh interregional, intersectoral and intergenerational consumption of this limited resource while staying within livable planetary boundaries for

humans today and allowing high-quality livelihoods for generations to come.

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

## Methods

### Geospatial analysis

In developing a map of potential sedimentary basins that would be acceptable for $CO_2$ storage, we focus on broad exclusions based on the geohazards, basin quality and potential engineering challenges. Mature sedimentary basins are the best candidates for $CO_2$ storage as they are the best understood and researched basins with the most available data. If they have been used for hydrocarbon exploration, some of the most crucial elements for $CO_2$ storage are already proven and in place, such as reservoir and sealing units with favourable petrophysical characteristics. Primary targets are stable, cold and under pressured basins. These types of basin are located mid-continent behind mountain ranges formed by plate collisions or at the edge of stable continental plates. Good examples in the Americas are the basins located behind the Rocky Mountains, Andes or Appalachian Mountains. In Europe, basins north of the Alps or west of the Ural and in Asia south of the Himalaya and south of Zagros Mountain chains are suitable candidates[20]. Present-day targets for $CO_2$ storage are depleted hydrocarbon fields or deep saline aquifers within geologically stable sedimentary basins, for example, the Sleipner Project on the Norwegian Continental Shelf[67]. However, one of the challenges in areas of extensive hydrocarbon exploration is the high number of abandoned wells. Although new drilled wells for the purpose of $CO_2$ injection will follow standards designed for this process, abandoned wells from the past exploration might be subject to well integrity failure during $CO_2$ injection and long-term storage. Possible risks are the appearance of fractures in the well cement matrix owing to chemical degradation under the influence of $CO_2$ or mechanically owing to increased reservoir pressure, which might lead to the escape of the stored $CO_2$ (ref. 68). A typical high-quality reservoir for $CO_2$ storage would have redundancy in the seal (layers of impermeable cap rock), favourable petrophysical parameters for injectivity and storage, sufficient depth, and low risk for reactivation of existing faults.

We start our geospatial analysis by first collecting the geospatial spread of the sedimentary deposits[69] and dividing them between onshore and offshore regions of interest at a global level using aggregated boundaries from the Global Administrative Area Project (GADM, v4.1, gadm.org). Several criteria must be considered when assessing the suitability of basins for $CO_2$ storage[28,29,70]. Basin characteristics such as storage capacity based on pore volume, depth of target formation, seal integrity, tectonic hazards and basin type are the most fundamental and non-negotiable criteria. Equally important is the assessment of existing resources of the basin, such as hydrocarbon or geothermal exploration, and groundwater with which $CO_2$ storage may compete. Finally, existing infrastructure, proximity to $CO_2$ emitters, and other socioeconomic factors such as local community acceptance for $CO_2$ storage may be critical. In our study, the selections for exclusion zones are made based on generalized global applicability of preventative boundaries for storage of underground $CO_2$ attributed to sedimentary basins and are categorized into three main groupings, which we discuss in detail further in this section.

Application of surface-level exclusion zones provide us with the suitable sedimentary available area metric, disaggregated by offshore and onshore geospatial attribution. To convert the suitable area into storage potential for $CO_2$, we utilized a modified volumetric storage calculation methodology, which assumes a lower level of storage potential per volume of sedimentary basin owing to limitation attributed to pressure increase and its effects on reduced injection rate within a closed system[71]. Here a closed system is defined by limitations on lateral flow owing to low permeability of basins and seals owing to faults. For this, we first converted the assessed suitable sedimentary deposit available area to volume using sedimentary depth mapping at 1° resolution[72]. Additional exclusion limits were then applied to the sedimentary depths to realize the effective assessed volume metric, which incorporated both areal exclusions and sedimentary depth exclusions. Conversion of assessed volumetric metric into storage potential was done with an assumption of 0.037 $GtCO_2$ storage potential per 1,000 $km^3$ of assessed sedimentary basin volume for a 50-year injectivity period including pressure-related injectivity considerations based on lower estimates in the available literature aligned with our precautionary assumptions[3,73].

Next, we briefly describe the type and reasoning for applying each exclusion layer. A summary of each layer's global impact on our estimation of a planetary limit is described in Supplementary Table 1, including a central, lower and upper sensitivity estimate ('Consistent storage assumptions across spatial scales and sensitivities'). Because each operation is computationally expensive, we apply exclusion layers sequentially. Thus, each estimate of volume removal is dependent on the previous layer applied. If each layer was applied only to the global technical potential, it would by definition exclude a larger volume than we estimated in our sequential application.

**Protected-area exclusion layer.** The first exclusion layer we apply is the policy based protected areas[74]. In this exclusion layer, we included areas that have the designation of protected areas as defined by the International Union for Conservation of Nature and the Convention on Biological Diversity. A comprehensive open-source dataset cataloguing the geospatial spread of the protected sites is developed under a joint project between International Union for Conservation of Nature and the United Nations Environmental Program. The database is entitled 'World Database on Protected Areas', which comprised 293,692 protected sites in May 2024 (Supplementary Fig. 1a). We overlayed this exclusion mask on our global dataset comprising onshore and offshore sedimentary deposit boundaries to demarcate sedimentary deposit areas outside of the protected zones.

**EEZ exclusion layer.** The second layer we apply is the global EEZs. The offshore territorial claim of a country is governed by the United Nations Convention on Law of the Sea. The conventions allow for 3 boundaries on the territorial claims of a country: (1) territorial sea, up to 12 nautical miles from the coastline; (2) contiguous zone, up to 24 nautical miles from the coastline; and (3) EEZ, up to 200 nautical miles from the coastline. Within the EEZ, the coastal state has sovereign rights of exploration, exploitation and management of natural resources in both the waters themselves and the seabed below.

States have rights regarding the protection and preservation of the marine environment in their EEZ, as well as the construction, operation and use of installations and structures at sea. In this study, we assumed that the maximum country-specific area attributable towards offshore sedimentary deposit will be within the boundaries of country's EEZ. Any offshore sedimentary deposit area outside of the EEZ will be considered international waters, hence unavailable for exploration and construction of deep-sea carbon injection infrastructure (Supplementary Fig. 1b). To mitigate the issues arising from overlapping claims over EEZ of a neighbouring coastal state, we have removed the EEZ area[75] where overlapping claims or a joint regime is present (Supplementary Fig. 1c,d).

**Polar-circle exclusion layer.** The third layer we apply is the area within polar circles. Polar circles, namely, the Arctic and Antarctic circles, demarcate the global circles of latitude at 66.5° N and 66.5° S, respectively (Supplementary Fig. 1e). In our study, we assume that the sedimentary basins north of the Arctic Circle and south of the Antarctic Circle do not contribute to the $CO_2$ storage potentials. This assumption is partly derived from the point of view of preserving polar ecosystems that are already sequestering large amount of $CO_2$ and partly owing to the intra-annual climatic and land-cover inaccessibility and associated increased costs of $CO_2$ storage.

**Seismic-hazard exclusion layer.** We next apply exclusions based on seismic-hazard zones. Basins in the proximity of active seismic zones,

for example, close to plate subduction areas, are less suitable for $CO_2$ storage owing to compromise of sealing units or the reactivation of existing faults during $CO_2$ injection[31]. The reactivation of existing fault structures in areas under far field stress poses a risk for leakage and for failure of these fault structures, potentially leading to low-intensity earthquakes. While in the hydrocarbon sector these geohazards are well understood, there remains uncertainty for the stability of geologic structures for the injections of reactive $CO_2$-rich water (for example, ref. 76). Future use of saline aquifers will therefore force future compromise between the risk of failure of seals that could cause leakage of $CO_2$. To approximate future choices to mitigate the risk of leakage, we assume that areas of moderate seismic hazard would be avoided. To incorporate such an exclusion zone, we use the global map of seismic hazard[77] to exclude areas that have a more than 10% of chance of peak ground acceleration breaching 0.115 g and 0.401 g (moderate in the US Geological Survey Instrumental Intensity scale[78]) in the next 50 years (Supplementary Fig. 2).

**Population-and-human-health exclusion layer.** Population centres have a crucial role in site suitability of carbon capture and storage facilities and form the basis of our next exclusion layer. Owing to the potential for toxicity mobilization, underground utility services provision, supply of fresh water, extensive human-made modifications to the topsoil and densely developed neighbourhoods preclude the global site suitability of carbon capture and storage facilities[79]. We utilize high-resolution datasets of built-up urban areas under different future population and urbanization scenarios[80] and choose our exclusion layer based on the urban area boundaries in 2100 based on Shared Socioeconomic Pathways narrative 5 (SSP5) entitled 'fossil-fuelled development' that incorporates high reliance on fossil fuels and CCS for future global growth[81]. In our central case, we assume a 25-km distance buffer around the urban boundary to account for a safety margin around the urban conglomerations. A sensitivity for this layer is also conducted by assuming a 5-km buffer zone to account for a less restrictive policy regime. Global maps of both the central case and sensitivity case, as well as a zoom-in to the Cairo metropolitan region is shown in Supplementary Fig. 3.

**Ocean-depth exclusion layer.** Here we assess exclusions related to the ocean-depth or water-depth boundary above the offshore basins. The offshore storage of $CO_2$ is advantageous as there is a very low risk of damage to infrastructure owing to induced earthquakes generated from the overpressure within the reservoir. Pumping in shallow seas will be relatively feasible, whereas in deep water the costs will increase. In the pre-salt basins offshore Brazil, drilling for hydrocarbons has been achieved down to water depths of 2,000 m (ref. 82). At the same time, some of the most environmentally damaging events in the history of oil and gas extraction have occurred due to extraction at depth, such as the Deepwater Horizon oil spill, which occurred at 1,500 m ocean depth in the Gulf of Mexico and was exacerbated owing to issues of reaching the well head[83]. In general, to achieve $CO_2$ storage within such deep-water reservoirs will, however, require a substantial shift in economic incentives, as the costs might be restrictively high.

To develop a prudent limit for offshore storage depth, we assessed >121,000 geospatial data points of discovered, abandoned, depleted and under-exploration offshore oil and gas wells[84–86] (Supplementary Fig. 4a–c). We then generated a histogram of water depth versus count of total offshore installations to derive cut-off for water-depth exclusion zones (Supplementary Fig. 4d). We find that about 70% of the installations globally are situated at a water depth of 100 m or less and about 95% are located in water of a depth of 500 m or less. The vast majority of oil and gas installations (>90%) are situated at water depths up to 300 m. Given these considerations, we use a central estimate for water-depth exclusion of 300 m with sensitivities of 100–500 m[87,88] (Supplementary Fig. 4d).

**Storage-depth exclusion layer.** Here we investigate the upper and lower cut-off for storage depth within the sedimentary deposit basins. Storage of $CO_2$ is ideally suited in aquifers below 800 m depth, beyond which the overpressure compresses the gas, its density decreases rapidly and $CO_2$ reaches a supercritical state, maximizing the storage volumes. Assuming a geothermal gradient of 25 °C km$^{-1}$ and 15 °C surface temperature, the $CO_2$ takes about 30-times less volume compared with surface conditions. Starting at 1.5 km depth, the density and occupied volume stay constant at about 36-times less volume compared with surface conditions[89]. On the basis of the sedimentary depth map for both onshore and offshore sedimentary basins, we estimate a mean depth of 1.9 km for onshore and 2.5 km for offshore basins for a 1 decimal degree global spatial areal grid (Supplementary Fig. 5a). A relatively small number of areal grid cells are present beyond 5 km sedimentary depth. Current literature (Supplementary Table 3) bound minimum and maximum estimates at 800 m and 4 km, respectively. We therefore apply a conservative depth requirement (minimum) of 1,000 m as a global mask to the database of sedimentary basins for our central case. For our central estimate, we apply absolute depth limit (maximum) to which $CO_2$ can be injected of 2.5 km to avoid bedrock and limit potential seismic activation of deep-rooted faults[30] based on the reviewed literature and following similar precautionary principles as for other exclusions. Additional sensitivity analysis is conducted for a range of minimum and maximum injection depths bounding our central estimate (Supplementary Table 2 and Supplementary Fig. 5b,c).

**Disputed-areas exclusion layer.** The territorial integrity of an important but relatively small number of maritime areas are at present in dispute, for example, in the South China Sea[90] (Supplementary Fig. 1d). Owing to the extreme uncertainty of potential carbon storage in these areas as well as those under war-like conditions, we exclude them from our analysis.

**Storage assumptions in transboundary basins.** It remains largely unclear how international treaties would perform in the future with multiple state actors injecting carbon into a common basin, either connected on land and especially at sea. The London Convention and the London Protocol has, so far, been the most proactive international body addressing carbon storage in subsea geologic formations. In 2006, the London Protocol parties adopted amendments to regulate subsea carbon storage, including introducing a risk assessment and management framework. In 2009, the parties amended London Protocol Article 6 on the export of wastes for dumping purposes, to enable parties to share transboundary subseabed geologic formations for $CO_2$ storage (LP.3(4)). However, this amendment has not entered into force, as the prerequisite two-thirds of the contracting parties have yet to ratify it. In 2019, parties to the London Protocol therefore adopted another resolution (LP.5(14)), which permits the provisional application of the amendment to Article 6, stipulating that two countries can now bilaterally agree to export and import $CO_2$ for subseabed geologic storage. To do so, they must deposit a formal declaration of provisional application with the Secretary-General of the International Maritime Organization and demonstrate they follow the guidance outlined in the London Protocol carbon storage risk assessment and management framework. However, as of January 2024, only 8 parties have done so[39]. We assume that parties would follow maritime law to allow transboundary usage based on common EEZs.

Common territorial boundaries on land require agreements between all parties sharing respective borders, and thus are not supported under a common international policy regime. Considering the uncertainties associated with the transboundary use of common sedimentary deposit basins and the country-level public perception towards CCS projects, we have assumed an exclusion buffer of 6 nautical miles along the terrestrial international boundaries in our central case to evaluate the scenario where international cooperation for shared storage

resource is not achieved in an effective manner. The choice for 6 nautical miles is derived from the definition of territorial waters boundary description within the United Nations Convention on Law of the Sea, which sets the territorial waters to be within 12 nautical miles. This assumption would effectively mean that sedimentary basin resources along the international boundary up to 11.1 km on both sides will not be used for injection and storage of $CO_2$, irrespective of the depth of storage and angle of approach to the storage site. We recognize that the pressure plume associated with $CO_2$ storage would reach farther than our assumed limit of 6 nautical miles, which is a limitation to our analysis should this consideration be built into future transboundary storage agreements. In addition to the central case, we include sensitivities of this exclusion layer to cover full 22.2 km (12 nautical miles) and no buffer considerations on both sides of the national boundaries (Supplementary Fig. 6).

**Consistent storage assumptions across spatial scales and sensitivities.** We compile global carbon-storage limits for a central estimate and sensitivity cases in Supplementary Table 1. We apply sensitivities using both numerical thresholds below or above a given central estimate and binary (included or excluded) thresholds (Supplementary Table 2), depending on the risk consideration. We begin Supplementary Table 1 with the total estimate of technical potential for carbon storage (see above). Each subsequent row presents the remaining storage available after excluding storage area based on the relevant risk consideration geospatial layer. Each exclusion layer is applied sequentially; thus the volumes reported depend on the order in which the layers are applied. When presenting the sensitivity performed for each risk consideration layer, we report the total storage considering only the sensitivity layer in question applied to the previous layer from the main calculation. We thus isolate the difference in storage volume owing to only the different calculation method for the layer in question.

Because our analysis framework is global and spatially explicit, we can summarize storage estimates at a country level. We combine all available storage per country administrative boundary for onshore storage and per country EEZ for offshore storage. We then perform the same calculation based on our final exclusion layer representing our assessment of the global planetary limit. We further identify existing sedimentary basins with current oil and gas infrastructure[84,87], which are prime candidates for initial exploration of $CO_2$ storage, and present their prudent storage potential. All values are summarized in Supplementary Table 5.

### Scenario analysis

We analyse scenarios assessed by the IPCC Working Group III, provided in its assessment database[91]. Scenarios are categorized by their global mean surface temperature characteristics by the IPCC, and we use the same categories with slightly clarified names in this paper as shown in Supplementary Table 6. We assess $CO_2$ emission in scenarios based on the variable 'Emissions|$CO_2$' and annual geologic carbon storage based on the variable 'Carbon Sequestration|CCS'. For all analyses, we take variable trajectories (based either on the IPCC R5 region classifications or global values) and interpolate any missing years using the 'interpolate' function Python library, pyam[92]. We calculate cumulative estimates of carbon sequestration from cumulative summing all yearly values until the year 2100, which corresponds to our estimate of total carbon storage in each scenario. We estimate the value of any given variable at the time of net-zero $CO_2$ emissions based on the first interpolated year at which $CO_2$ emissions cross from a positive value to a negative value for each scenario.

To estimate carbon-storage exceedance at net-zero levels, we calculate the remaining carbon-storage volume at the time of net-zero $CO_2$ emissions in each scenario and divide by the level of yearly geologic $CO_2$ injection at that point, resulting in the number of years of storage remaining if injection levels were held constant at these levels. We

estimate carbon-storage needs in a given scenario after its time horizon by using a first-order spline interpolation extrapolating beyond the model horizon until 2300 (so-called slinear interpolation in the Python library scipy). We then estimate the resulting cumulative carbon-storage trajectories from these trajectories and identify at which point they cross specified thresholds.

All calculations are provided in an open-source GitHub repository at https://github.com/gidden/2024_gidden_cstorage.

**Regional scenario analysis.** IPCC scenarios provided data beyond global levels at the scale of five macroregions. Countries are allocated to each macroregion as described in Supplementary Table 7. We analysed the global carbon-storage thresholds as shown in Fig. 3 also for every IPCC macroregion in Supplementary Figs. 8–12. In particular, the R5ASIA region sees levels of carbon storage beyond the regional boundary consistent with our assessed global planetary limit even in the modelled time horizon up to 2100.

## Data availability

The data generated in this study, including country-resolves storage estimates, are available in Supplementary Information and on Zenodo at https://zenodo.org/records/15657543 (ref. 93). The workflow for data generation was executed on ArcGIS Pro software using various spatial datasets and exclusion zone layers, which are cited in the study[69,72,74,75,77,80,84–86,90]. National boundaries were derived using GADM project, version 4.1 (https://gadm.org/data.html). All calculations are provided in an open-source GitHub repository at https://github.com/gidden/2024_gidden_cstorage. Data generated in this study can be explored further at https://cdr.apps.ece.iiasa.ac.at/story/prudent-carbon-storage.

## Code availability

All code used to generate results can be found at https://github.com/gidden/2024_gidden_cstorage.

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

**Acknowledgements** We acknowledge the funds received from the ERC-2020-SyG GENIE grant of the European Union, grant number 951542, from the European Union's Horizon Europe research and innovation programme under grant agreement number 101081521- UPTAKE - Bridging current knowledge gaps to enable the UPTAKE of carbon dioxide, and the Korea Environmental Industry and Technology Institute (KEITI) through the Climate Change R&D Project for New Climate Regime, funded by the Korea Ministry of Environment (MOE) (RS-2022-KE002096). M.B. acknowledges financial support from the ASMASYS project, funded by the German Federal Ministry of Education and Research (BMBF), grant number 03F0898E. A.C.K. acknowledges the financial support from the Portuguese Fundação para a Ciência e Tecnologia (FCT, I.P.MCTES) individual fellowships (https://doi.org/10.54499/2023.07816.CEECIND/CP2831/CT0007) and through national funds (PIDDAC) (UID/50019/2025 and LA/P/0068/2020 https://doi.org/10.54499/LA/P/0068/2020), and the Horizon Europe research and innovation programmes under grant agreement number 101081661 (WorldTrans). M.B. acknowledges the assistance of R. Majewski and L. Oechtering in the compilation of information on the CCS policy landscape. We thank C. Bergero who compiled research data on carbon injection rates. We thank M. E. Maes for her support in figure preparation. M.J.G. is also affiliated with Pacific Northwest National Laboratory, which did not provide specific support for this paper.

**Author contributions** M.J.G. conceived of the research and wrote the first draft of the paper. Analysis was performed by S.J., M.J.G., J.J.A., A.-B.C. and J.R. S.J. and M.J.G. created all figures. M.J.G., S.J., J.J.A., A.-B.C., M.B., E.B., A.C.K., K.R., H.J.S., C.-F.S. and J.R. contributed to drafting, editing and reviewing the paper.

**Competing interests** The authors declare no competing interests.

**Additional information**
**Correspondence and requests for materials** should be addressed to Matthew J. Gidden.
