## [Peer Review file · Nature]

A prudent planetary limit for geologic carbon storage

Corresponding Author: Dr Matthew Gidden

Version 0:

Reviewer comments:

Referee #1

(Remarks to the Author)

This paper extends the concept of planetary boundaries to the amount of geological storage available. In principle, this could be a valuable contribution to the literature if framed in a way that focuses on the risk of needing to rely so heavily on carbon dioxide removal as a climate stabilization method rather than the potential short-comings of geological storage capacity. However, as written, I have three major concerns with the paper.

First, there is little that is novel in the paper that has not been discussed in previous literature, including much of the work going back to the IPCC special report on Carbon Capture and Storage, and extensive work by authoritative organizations such as the U.S. Geological Survey. It has long been recognized that not all the geological storage "resource" would be available due to economic, environmental, and social concerns. The methodology of using exclusion layers is also not novel, and in fact, a standard approach for assessing capacity in basins.

Second, some of the "boundaries" are overly stringent. For example, excluding oil and gas resources on the basis that they are used for EOR does not make sense. The ratio of CO₂ to oil recovery can be adjusted so that more CO₂ is stored per barrel of oil than would be released to the atmosphere when it is burned. Furthermore, in a world of declining demand for oil, depleted fields may provide a low cost and secure option for CO₂ storage. In this case, the biggest (and significant) risk is that old wells leak CO₂ to shallower depths, and even possibly to the surface. Nevertheless, excluding depleted oil and gas reservoirs is not scientifically justified on the basis that is currently used with EOR. The cut-off depth of 2,500 m depth is also arbitrary. There are many places where the basement rock is far greater than a depth 2,500 m and therefore, risks of seismic activity associated with basement rocks not a significant concern. Similarly, constraining water depth to 300 m is also arbitrary. Off-shore rigs are available today that can drill in much deeper water than this cut-off. If some or all these limitations are relaxed, CO₂ storage capacity could easily double, bringing into question the conclusion that "this carbon storage potential limit sets a cap of about 0.7°C on the total warming that can ever be sustainably reversed if the full prudent potential is used for CO₂ removal."

My third concern stems from the framing of the paper which suggests that the capacity will be far less than is needed, rather than addressing the more important point that relying too much on carbon dioxide removal for climate stabilization is a risky strategy. With the "planetary boundary capacity" of 1,500 Gt, that would be sufficient to capture and store 15 Gt per year (300 times more than the 50 Mt per year of geological storage today) for 100 years. If the "planetary boundary" capacity is double this, which is likely based on the above discussion, the "cooling potential" would double to 1.4 oC, enabling an enormous 30 Gt/year (equivalent to 83% of today's fossil fuel emissions) for 100 years. Accomplishing this degree of scaleup and paying the associated costs for carbon dioxide removal will be an enormous challenge in and of itself. In light of this, the risk of not accessing this storage capacity due to financial limitations and lack of political will, are a far greater than the risk of not having the storage capacity we need. As written, this paper is most likely to have the effect of further discouraging use of the capacity we do have, particularly in the short run, for reducing emissions from industrial sources such as cement, steel, ammonia, where few other options exist today.

(Remarks on code availability)

Referee #2

(Remarks to the Author)

General comments:

Using geospatial analysis combined with various risk exclusion layers the authors estimate a “prudent” boundary for CO₂ storage in sedimentary basins. They estimate that the prudent potential is around 1,500 GtCO₂, a value that could well be exceeded this century in emission scenarios that rely heavily on CO₂ storage to compensate for residual emissions and to achieve net negative CO₂ emissions.

This is the first study that presents an estimate of geological storage that goes beyond technical potential. It is an important contribution as it changes the narrative from CO₂ storage being an unlimited resource. Rather, it shows that prudent CO₂ storage is limited, implying that it needs to be used strategically to satisfy various needs. The study is topical as remaining emission scenarios that are consistent with limiting warming to the Paris Agreement climate goal require carbon dioxide removal and storage, with geologic storage being the only permanent solution.

The methodology is straightforward and clearly explained. My main reservation is the lack of uncertainty analysis – the authors present a best estimate of the prudent storage potential without an uncertainty range. It is hard to believe that there is no uncertainty in the estimation of storage potential in sedimentary basins and the data used in the various risk exclusion layers.

The study is logically structured and well written but could benefit from further elaboration and clear definitions in some instances. Figures could be used to more clearly illustrate the points described in the text – figure references are not always obvious.

Specific comments:

I. 21: “We establish a prudent planetary boundary of around 1500 Gt of CO₂ storage”. What is the uncertainty for this estimate?

I. 24: “Can at most reduce global temperature by 0.4-0.7°C”. This is a highly optimistic estimate. As the authors mention in the manuscript, the temperature reduction is likely smaller due to inertia effects and asymmetries in the climate-carbon cycle response.

I. 41: Define CCS.

I. 56-59: Clarify that CDR with biological storage is not permanent (i.e. storage timescale is less than several hundred to a thousand years).

I. 177: “physical storage potential of 11,784 GtCO₂”. What is the uncertainty around this estimate?

I. 181-187: “Many countries...”: The points described in the text cannot easily be seen in Fig. 2. I suggest to add a panel that shows the total (onshore+offshore) prudent potential for countries.

I. 187-189: If I read Fig. 2c correctly, it suggests that the largest restriction is from maximum storage depth, not Arctic storage.

I. 194-197: Suggest to rewrite in less technical language.

I. 217: Unclear how the bullets relate to Fig. 3a.

I. 231 (and elsewhere): Clarify what percentages (33%, 50%) refer to.

I. 269, 277: The calculation assumes that the TCRE is the same for CO₂ emissions and removals, which is not the case due to inertia effects and asymmetries in the climate-carbon cycle response to CO₂ emissions and removals. The authors allude to this asymmetry later in the manuscript (I. 343-347), but the implications of this asymmetry should already be discussed here. An additional relevant reference is Zickfeld et al., 2016.

I. 323: Is “preventative” the same as “prudent”? Suggest to use consistent terminology.

I. 343: See additional reference provided in earlier comment.

Fig. 2: i) Some country ISO codes are not obvious. Suggest to use country names in radial diagrams. ii) Clarify what unit the technical storage potential is given in in the radial diagrams (numbers in parentheses) iii) consider including a panel that shows total (onshore + offshore) storage potential, which is referenced in the text. To make room the radial diagrams could be removed – Fig. 2d illustrates countries’ storage loss in a more intuitive way.

Fig. 3: Clarify what is meant by “preventative limit” in the legends. Is it the same as the prudent planetary boundary? Clarify what percentages (33%, 50%) mean.

Fig. 4: Units are missing from horizontal axes.

References:

Zickfeld, K., A.H. MacDougall and H.D. Matthews, 2016, On the proportionality between global temperature change and cumulative CO₂ emissions during periods of net negative CO₂ emissions, *Environmental Research Letters*, 11, 055006. <https://iopscience.iop.org/article/10.1088/1748-9326/11/5/055006>

(Remarks on code availability)

Referee #3

(Remarks to the Author)

The manuscript by Gidden et al. present a global CO₂ storage capacity assessment based on different geological, social, environmental and regulatory/legal aspects that produce a prudent (as in not-so-optimistic) storage potential. The authors use this new assessment to constrain the capacity of CO₂ storage in achieving climate targets. The paper is well written and, to the best of my knowledge, the implementation of the capacity assessment within the IPCC scenarios is not an issue. My review focuses mainly in the aspects regarding the analysis of CO₂ storage potential, as it is closer to my field of expertise.

I have a number of concerns regarding the authors' assessment of CO₂ storage potential. Please find also attached the manuscript with further comments and questions.

- My main concern is that the authors have not included the effect of the injectivity in their assessment, probably the most important factor beyond volumetric capacity in terms of accessibility to storage resources. There is a high uncertainty in the estimation of injectivity values, because it is necessary to have a good knowledge of the subsurface (something that can be difficult/expensive, especially in hydrocarbon-poor provinces), but this information can really impact the calculations reducing the overall optimistic volumetric capacity calculations. The risk of reduced injectivity can probably be more damaging to the development of CCS than other factors mentioned by the authors (e.g. the risk of induced seismicity); low injectivity-related issues have impacted CO₂ storage projects before, and have even led to shutting down entire projects (see the discussion in Huerta et al., 2020). The lack of robust injectivity information was already flagged in the work by Lane et al. (2021), which point out that "deep uncertainty over the sustainable injection rate at any given location will constrain the pace and scale of carbon capture and storage deployment". Grant et al (2021) further discuss this issue, and urge for the inclusion of injection rates in the capacity estimations – and the subsequent IAMs to which they input – and I strongly agree with this view. Ringrose and Meckel (2019) already attempted to produce a global assessment considering injectivity, where they derived generalised pressure-depth functions to produce a proxy for injectivity. Lane et al (2021) noted that these type of injectivity analyses can also be misrepresentative, because injectivity may decay with time, but you could incorporate a similar assessment to approach this issue. IMHO, the injectivity aspect is crucial to actually produce a more reliable high-level capacity estimate and create the impact that the authors are looking for. Without it, I do not see that extra relevance of the analysis made, especially considering the arbitrariness in some of the criteria used in the capacity assessment.
- The authors claim that "to date, no consistent estimate of global carbon storage potential has included these various factors" (line 85 in the manuscript), but there are plenty of such capacity assessments in the literature, perhaps not applying the exact same criteria, but with similar approaches. See for example the dedicated IEA analysis (Malischeck and McCulloch, 2021), the study by Wei et al. (2021) or the references in Zhang et al (2024). In that sense, the authors should discuss what are the main differences between their methodology and the ones already published, and to analyse the differences between the storage capacities calculated using their assessment and the other examples in literature. I think making this comparison it would be useful to put your assessment into perspective.
- In general, the authors mix absolute criteria (e.g. inside or outside protected areas, 0-1) with more interpretative criteria. The water depth is a clear example of this mixing, as it is currently set at 300m – 1500m depth, but the authors fail to offer a convincing reason to why a platform located at 301m or 1501m should be disregarded. Water depth in itself is not necessarily an excluding factor (e.g. in opposition to the lack of a competent seal), it may "just" make the whole project more expensive (it may make it too expensive or risky, but that is a case-specific issue). There are other factors that can make a project expensive/riskier (and also too expensive/risky) that are not included in the analysis, for example the depth of the reservoir (the deepest cut off it is currently only considered from a safety point of view) or the distance from the CO₂ sources (e.g. longer routes may lead to more incidents, plus increased transportation costs). I suggest that the modelling should focus on absolute (as in binary criteria, qualifying or disqualifying) factors, and leave out other more interpretative (or case specific) factors. If the aim is to provide absolute thresholds/range of storage capacity potential (as in a screening exercise), then perhaps it would be more appropriate to use the entire range in relative criteria and afterwards use the sensitivity to include the authors' criteria (i.e. to give some weight to the analysis).
- Storage in transboundary basins seems to me another arbitrary factor to be considered. Indeed, there can be issues associated with transboundary storage (same as with other types of activities), but there are also opportunities for collaboration and even cost reduction (and hence reduced risk) in transboundary zones, a clear example being the North Sea, where transport and storage shared across countries is the key to the development of the Northern Lights project (<https://www.equinor.com/news/20220829-important-step-to-decarbonise-europe>). Having issues associated to the operation in transboundary regions is thus not a direct disqualifying criterion, in my opinion. Also, there are other important factors that are not considered in the analysis; for example, if CCS is banned in certain country is definitely a more

restraining aspect, and potentially with a stronger effect in the capacity assessment than the transboundary issues. Another aspect not considered within this criterion is the interaction between the zones of influence of the storage (e.g. the pressure plume), which can sometimes extend tens of kilometres beyond the plume of CO₂; it is more likely that storage operation can perturb the pressure state in a contiguous reservoir than the plume of CO₂ trespassing the boundary. The 22.2 km is the distance from the shore to the international waters, but this is not necessarily a limitation for economic activities in the vicinity of these boundaries (e.g. fisheries or wind farms) so why would they be a limitation for CCS?

Other minor issues (some are also highlighted in my comments within the manuscript):

- This is perhaps a language issue, but the authors use “carbon” and “CO₂” interchangeably when talking about geological storage. I am not a native English speaker, but I would be careful in the usage of these terms; I understand that “carbon” is broadly used in the climate/environmental field, but the only type of carbon that is stored in CCS is carbon dioxide. Because of this lack of consistency in the terms used, it is sometimes difficult to discern to which one are the authors referring throughout the text. For example, when referring to the planetary boundary concept introduced in line 106, does it refer to carbon in general (as in the C cycle) or to CO₂ in particular? The authors should be consistent throughout the manuscript to avoid misinterpretations.

- The authors report global capacity estimations of 10,000-40,000 Gt CO₂ from literature, with industry estimates of around 14,000 Gt CO₂, but their starting capacity is on the lower end of this range (11.784 Gt CO₂). The calculation of this initial geological potential should be better described in the Methods. Does the sedimentary basin data analysed already include the depths? Also, the authors should delve onto this discrepancy, which clearly impact the final assessment.

References:

Grant, N., Gambhir, A., Mittal, S., Greig, C., & Köberle, A. C. (2022). Enhancing the realism of decarbonisation scenarios with practicable regional constraints on CO₂ storage capacity. *International Journal of Greenhouse Gas Control*, 120, 103766.

Huerta, N. J., Cantrell, K. J., White, S. K., & Brown, C. F. (2020). Hydraulic fracturing to enhance injectivity and storage capacity of CO₂ storage reservoirs: Benefits and risks. *International Journal of Greenhouse Gas Control*, 100, 103105.

Lane, J., Greig, C., & Garnett, A. (2021). Uncertain storage prospects create a conundrum for carbon capture and storage ambitions. *Nature Climate Change*, 11(11), 925-936.

Malischek, R. & McCulloch, S. - IEA (2021), The world has vast capacity to store CO₂: Net zero means we'll need it, IEA, Paris <https://www.iea.org/commentaries/the-world-has-vast-capacity-to-store-co2-net-zero-means-we-ll-need-it>

Ringrose, P. S., & Meckel, T. A. (2019). Maturing global CO₂ storage resources on offshore continental margins to achieve 2DS emissions reductions. *Scientific reports*, 9(1), 1-10.

Wei, Y. M., Kang, J. N., Liu, L. C., Li, Q., Wang, P. T., Hou, J. J., ... & Yu, B. (2021). A proposed global layout of carbon capture and storage in line with a 2 C climate target. *Nature Climate Change*, 11(2), 112-118.

Zhang, Y., Jackson, C., & Krevor, S. (2024). The feasibility of reaching gigatonne scale CO₂ storage by mid-century. *Nature Communications*, 15(1), 6913.

(Remarks on code availability)

Referee #4

(Remarks to the Author)

This paper seeks to establish a framework for a planetary boundary for geological CO₂ storage. It does so by performing a spatially explicit analysis that excludes areas based on depth, active seismic zones, protected and polar areas, areas of human settlement, and transboundary basins. The paper will be of wide interest. It is novel, and uses data appropriately, employing a useful methodological approach. The only issues I would identify relate to the presentation, clarity, and context.

There is one issue that should be addressed, regarding the clarity of the introduction, and a second issue that I would invite the authors to reflect on, regarding the suitability of the planetary boundaries framework.

1. The introduction should be reworked for clarity around the basic concepts and activities being described.

Both the abstract and the first few paragraphs have some conflation between negative emissions / CDR and geologic storage / CCS. For example, the abstract begins with three sentences about geologic storage; then the fourth sentence talks about both near-term emissions reductions (including CCS, or not?) as well as “durable carbon dioxide removal”, which should probably be “carbon dioxide removal that uses geological CO₂ storage” for accuracy.

It will be critical to be very clear about the distinction up front, and throughout the paper. It is brought up somewhat in the third paragraph, but still a bit confusing for readers (and the third paragraph actually works better as the second paragraph — because the first paragraph is about expectations, and so is the third — better not to tack back and forth). The fix is as simple as having a sentence along the lines of “Geologic storage is necessary for many CDR approaches, such as DACCS and BECCS” — but also just having clear basic definitions of CDR, CCS, and storage. It will also be important to define “durable” CDR as being not ocean alkalinity enhancement or enhanced weathering (I guess — although if a direct ocean capture company is able to successfully make solid carbonates, wouldn't that also be durable CDR, albeit without CO₂ injection into

sedimentary rock?)

In lines 61 – 68, it is again rather unclear if the authors are talking about CCS or geologic storage. The first sentence reads as if CCS for point-source / mitigation, the rest of the paragraph seems to talk about geologic storage for CDR. And the reader doesn't really have a sense of what percentage of storage is for point-source capture vs. CDR in the scenarios being discussed. It would be really nice to add a figure illustrating that context for people (i.e., in most scenarios, is more storage from point-source mitigation or from CDR, and how does that change over time)? Do the takeaways here point towards not just avoiding excessive CDR, but to avoiding fossil gas or industrial decarb strategies with CCS? The latter is not really present in the paper. Given that we are unlikely to even do much CDR, the takeaways of this analysis for industrial decarbonization in sectors like steel or hydrogen could be better foregrounded. You could imagine a version of the paper that was centered on the need to better incentivize green hydrogen vs. blue hydrogen, for example.

The authors might also want to clarify what the literature says about the potentials of non-geologic-storage CDR approaches. These findings could be read in two ways that the authors do not mention. One, the findings could be read as implying the need for CDR that doesn't involve geologic storage, such as ocean alkalinity enhancement or enhanced weathering or ocean fertilization. Two, there is a reading that 1.5°C is even more unachievable without solar radiation modification. You may even end up with people resurrecting that idea of just injecting CO₂ into the deep ocean, if you define all these onshore areas as too risky. I am not suggesting tackling all that in the paper, but it would be helpful for readers to acknowledge that there are more methods of CDR that don't hinge on geologic storage. In line 58, it is mentioned that the "forestry-construction pump" would reduce dependence on geologic storage — why mention that one (I'm skeptical about this being of climate-significant scale), and not the others that people are currently enthusiastic about?

2. Is "planetary boundaries" the right framework for this paper? What is gained by this, vs. framing it just as a right-sizing estimates of storage using a holistic risk assessment, or "a more realistic view of storage capacity given social and geologic factors", or a frame around "equitable division of storage reserves and the need for planning", etc?

There are two main issues with employing this framework.

Issue 1 - As I read it, the "framework establishes global boundaries, which, if transgressed, substantially increase the risk of a change in Earth system functioning and/or its long-term state and trajectory" (from Rockstrom et al. 2024 review article, 'Planetary Boundaries guide humanity's future on Earth', in Nature Reviews Earth & Environment). So this seems like the framework is focused on the boundaries that risk the change in Earth system functioning, i.e. breaching the boundaries poses a global risk. But it seems like with geologic storage, doing more of it and breaching the boundary poses a bunch of local risks? I believe this has been a longstanding critique of the planetary boundaries framework — "its inclusion of subglobal-scale, nonsystemic, and aggregate processes that have shown little evidence of threshold behavior so far", as Biermann and Kim (2020) identified in their ARES review article on the "boundaries of the planetary boundary framework". I know that the authors related to planetary boundaries have also stated that no global thresholds are assumed within the framework, but it still seems like a point of confusion, and I wouldn't expect all the readers of the paper to be familiar with the nuances and debates around the framework.

Issue 2 - It seems that another one of the main critiques of the planetary boundaries framework is that the setting of boundaries is arbitrary, and more of a political exercise or social construction rather than something derived from the scientific literature on thresholds. This relates to what Biermann and Kim (2020) identify as the critique of boundaries derived from Science and Technology Studies — that "boundaries are proposed by experts without stakeholder consultation or public participation and deliberation." I mean, the parameters in the study are the authors expert opinion, not developed through deliberation.

In short, the "planetary boundaries" idea suggests a biophysical limit, but the parameters here include social judgements of risk. For example, in lines 82 – 98 — this paragraph moves from talking about perceptions of harm and risk to talking about planetary limits, and then in line 106-107, the definition is explicitly about harm prevention and risk avoidance. So this reads as basically a social construction and not a biophysical limit. Doesn't this kind of undo the discursive power of the planetary boundaries framework (that it relies on science)?

Whether or not a person wants to live within 25 km of a storage well can be a matter of social circumstance, but it can also be a social judgement based on one's values. I have met people in places like Texas who live literally next door to CO₂ injection wells that they can see from their yard and were not troubled by this. Other people may have no practical choice about where to live due to income and other factors, but still, what that buffer should be is a question for society. While public opinion of CO₂ storage is low today, future people may prefer CO₂ storage to alternatives of warming or solar geoengineering, if those end up being the choices on offer. Similarly, whether or not it is anticipated that nation-states can agree on managing a basin is a governance matter rather than a biophysical one. The choice about the Arctic circle is also a social choice; e.g. there may be actors in Alaska who are scoping storage above the Arctic circle and see that as desirable and workable. We already know Santos is interested in CO₂ storage on the North Slope with the Pikka project, and ConocoPhillips is also scoping North Slope sequestration. So the value choices of these authors might not be the same as companies or even Alaskan voters, potentially. Same thing for seismic hazards — there are already areas in S2 fig (a) (note typos on this figure) that are being explored or developed for storage. It seems that regulators think they can manage this through things like the Class VI framework in the US. The authors may differ in their opinion, but society may choose something else.

I think that critics of the planetary boundaries framework would read this paper and say, see, here's one more piece of evidence that planetary boundaries are just a made up expert construction and not grounded in biophysical reality. Do the authors want to arm the planetary boundary critics in that way?

Personally, I would find the paper more convincing without this framing. It is already interesting in terms of providing a more realistic assessment of CO₂ storage potential than is indicated in the literature, and highlighting the tradeoffs between using storage for CCS vs. CDR. To me, mixing it up with the planetary boundaries framework just confuses the issue. People are already critical of scientists for using the language and institutions of science to defend their value judgements and personal preferences. I would give any framing that might appear that way a wide berth in these times, when public trust in scientific neutrality is paramount.

(Remarks on code availability)

Version 1:

Reviewer comments:

Referee #2

(Remarks to the Author)

The authors clearly put considerable effort into revising the manuscript. However, one of my main comments was not addressed to my full satisfaction. The authors considered uncertainties regarding the criteria used in the exclusions layers in their analysis but missed to report and discuss the resulting range in the prudent planetary limit (1,285 to 2,710 GtCO₂; Table S2) in the main manuscript. Given that relaxing constraints (within uncertainty bounds) on some of the exclusion criteria nearly doubles the prudent storage limit, acknowledging these uncertainties in the context of storage requirements in mitigation scenarios appears warranted.

Minor comments:

l. 21, l. 191, l. 286, Fig. 2b: Report full uncertainty range for the prudent planetary storage limit.

l. 288, "based on assessed uncertainties": clarify that you are referring to uncertainties in exclusion criteria.

l. 290-291, l. 299: In precise terms this quantity is referred to as the "transient climate response" to cumulative CO₂ emissions.

l. 292; "similar" should read "similar in magnitude".

Table S2: Include in extended data file for ease of access.

(Remarks on code availability)

Referee #3

(Remarks to the Author)

I have read with interest the revised version of the manuscript submitted by the authors, as well as their detailed responses to the reviewers' feedback. I appreciate the effort they have made to address each of the points raised and to incorporate our comments into this new version.

The revised manuscript introduces two significant changes. First, the authors have refined their capacity analysis based on some of the feedback provided, making the estimates more accurate and realistic. Second, they have removed the rationale based on planetary boundaries, which—like Reviewer #4—I found unconvincing and not particularly helpful in highlighting the strengths of the study. Furthermore, the authors have developed a country-by-country capacity assessment and intend to make the associated database openly available to the research community, which I believe will be a valuable resource and welcome this initiative.

However, I remain concerned about two key aspects that, in my view, considerably reduce the impact of the work. The first relates to the novelty of the study, which both I and Reviewer #1 criticised in our initial reports. In their response, the authors appear to acknowledge this limitation. For example, they state that "our goal here is not to claim novelty in how we define our carbon storage limits, but to apply them in an open and documented manner to stress how they will limit CCS plans." Once again, the argument that storage capacity may represent a limiting factor in the deployment of CCS is not new and has already been articulated in previous literature, including, for instance, the work by Lane et al. on injectivity as a limiting constraint.

The authors also claim that "there is a strand of existing literature... that is rarely consistent in how the total volume of storage capacity is defined." However, as I pointed out in my earlier review, the absence of consistency across studies does not imply the absence of valid methodologies, and one could equally argue that the analysis presented in this manuscript may also lack consistency. In my view, it is essential to place the proposed analysis in context, allowing for meaningful

comparison with existing estimates, rather than simply highlighting how it differs. Given the number of assumptions involved, it is understandable that the study cannot reflect the full complexity of the system. Nevertheless, the deviations observed when comparing these results with more detailed assessments are considerable (e.g. the study assigns a storage potential to the USA that is approximately three times lower than the estimate provided by the USGS). Such discrepancies—though perhaps justifiable due to the simplifying assumptions—inevitably raise questions about the robustness of the final results. Finally, I am not fully convinced by the authors' rationale for excluding injectivity from their analysis, beyond the assertion that "more explicit consideration of injection feasibility would likely reduce our estimate of usable storage potential." Given the magnitude of the discrepancies between their volume estimates and those reported in the literature, and the fact that injectivity is not accounted for, I find the overall value of the proposed study to be limited. That said, I wish to reiterate that I do consider the work to be useful, and I do not intend to undermine the efforts of the authors, which I recognise and respect.

(Remarks on code availability)

Referee #4

(Remarks to the Author)

I appreciated the reframing in this paper. The authors write in their response, "We feel that it is important to highlight that what we have developed here is an analytical framework and analysis to holistically assess geological, technical, and risk-based limits to carbon storage and hope that this is an acceptable compromise in the framing of this approach." This is a clear distillation of what the article offers, and it comes through in the revised piece. The planetary boundaries conceptual framing risked muddling this useful offering.

The revision also addresses some of my concerns about the clarity of the distinctions being made between CCS and CDR. Very minor point, I'm not sure that storing carbon in minerals through enhanced rock weathering would be "less durable" (line 65), though it would be harder to measure. On a writing level, I am glad to see the argument mentioned up front in the introduction.

I also agree with the authors about the novelty of the paper; it is important for the field to have additional analysis that moves beyond purely technical potential of geological storage capacity.

Figure S7 is useful context for readers.

I also found the introduction of the uncertainty analysis and its discussion in the response to reviewers document useful and convincing.

(Remarks on code availability)

Response to Reviewers for:

A prudent planetary limit for geologic carbon storage

Authors: Matthew J. Gidden^{1,*}, Siddharth Joshi¹, John J. Armitage², Alina-Berenice Christ², Miranda Boettcher^{3,4}, Elina Brutschin¹, Alexandre C. Köberle^{5,6}, Keywan Riahi¹, Hans Joachim Schellnhuber¹, Carl-Friedrich Schleussner^{1,7}, Joeri Rogelj^{1,8}

Affiliations:

¹ International Institute for Applied Systems Analysis, Laxenburg, Austria

² IFP Energies nouvelles, Rueil-Malmaison, France

³ German Institute for International & Security Affairs (SWP), Berlin, Germany

⁴ Copernicus Institute of Sustainable Development, Utrecht University, Utrecht, the Netherlands

⁵ Instituto Dom Luiz (IDL), Faculdade de Ciências, Universidade de Lisboa, Lisboa, Portugal

⁶ Potsdam Institute for Climate Impact Research, Potsdam, Germany

⁷ Geography Department and IRITHEsYS Institute, Humboldt-Universität zu Berlin, Berlin, Germany

⁸ Centre for Environmental Policy and Grantham Institute – Climate Change and Environment, Imperial College London, London, UK

* Corresponding author: gidden@iiasa.ac.at

Overview

The authors would like to heartily thank all four reviewers for their time and effort providing very thorough reviews of our manuscript. We believe our manuscript has greatly improved by deliberately considering their review comments and adjusting our analysis, framing, presentation, and discussion accordingly.

We have striven to address each point raised in sufficient detail and believe we have addressed each reviewer's concerns to the best of our ability given the scope of the paper and analytical approaches available to us. We have completely reassessed the geospatial analysis of global carbon storage in sedimentary basins with multiple areas of uncertainty across risk dimensions. We now provide a structured assessment of variation across assessed dimensions and have produced an open-access dataset at country level which allows for systematic exploration across each sampled uncertainty dimension. We present in the supplemental information our global estimate with uncertainty quantification and the central estimate per country across multiple considerations. The country dataset fully exploring all uncertainty dimensions is now being submitted to Scientific Data, as we felt it was sufficiently rich and detailed to merit additional description.

In our following response, we provide direct responses to each reviewer comment but first address thematic concerns raised by reviewers in a consolidated manner across three primary areas:

1. Issues related to uncertainty analysis and key analytical assumptions
2. Issues related to subsurface injectivity
3. Issues related to framing our analysis around the planetary boundary concept

Response related to uncertainty and analysis assumptions

We have greatly expanded our analysis to include key uncertainties in the most important exclusion dimensions we explore. In particular, we focus on those dimensions highlighted by reviewers as seeming arbitrary. We note that any analysis that seeks to perform a spatially explicit global assessment of geologic carbon storage must make some simplifying assumptions, and we have striven to better justify ours. At the same time, we agree wholeheartedly with the reviewers that a more structured uncertainty assessment is appropriate.

In particular, our limitation of water depth and maximum injection depth were highlighted by reviewers as needing further attention. We addressed each of these in the most appropriate manner we could, given our resources and expertise.

To solidify an assumption regarding maximum ocean depth which countries can safely assume will be exploitable for mitigation strategy planning, we look to existing oil and gas infrastructure which is currently economically profitable. We assume that over the course of the planning horizon for countries that even the most ambitious countries would enact, through policy or other mechanisms, carbon price structures that would result in similar exploration and storage efforts as is currently ongoing in the oil and gas industry. To test this assumption and provide uncertainty bounds for our analysis, we apply sensitivities at depths consistent with most oil and gas infrastructure installations and going beyond the depths at which the vast majority of installations operate. The new geospatial analysis of existing infrastructure and their respective operating depths as well as the sensitivity threshold and descriptions are provided in our methods section 1.6 Ocean Depth Exclusion Layer and associated sections in the Extended Data.

We also explore and expand our assumption on maximum storage depth, which was also criticized by certain reviewers. To provide a more robust assessment, we performed an extensive literature review to better understand other assumptions. We were able to find 11 different sources largely in the academic literature which report estimates of minimum and maximum storage depth, which is now reported in SI Table 3. The majority of the literature we assessed finds that maximal storage depth ranges from around 800-3000m. One study claimed storage depth possible in the gulf of

Mexico up to 3500m, but noted that it was unclear about this range due to either pressure in the geopressure zone equilibrating with fracture pressure or loss of permeability. The maximum storage depth we could find was from the USGS which uses a boundary of 3962m based on compression requirements. Taken together, and given the large preponderance of the scientific literature, we maintain a central estimate for maximum storage depth of 2500m, but apply a range between 800m and 3500m in our primary analysis to acknowledge and show the uncertainty in this key parameter in our reported results.

We then reframe key figures and outcomes in our manuscript, e.g., around total temperature drawdown potential, in terms of this more robust uncertainty analysis. For example, our abstract now includes: “Fully using geologic storage for carbon removal caps possible global temperature reduction to 0.4-0.7°C (0.35-1.2°C).”

Response related to issues around injectivity

We appreciate and recognize the Reviewers’ comments related to injectivity limits (be they geophysical or techno-economical) being a critical factor when holistically assessing the feasibility of a future with large-scale CCS. There is a strong and growing literature both assessing the feasibility of large-scale CCS scale-up from a political and technological perspective as well as a burgeoning literature critiquing scale-up rates in scenarios. Indeed, we cited recent portions of this literature already throughout our manuscript (e.g., Lane et al 2021, Grant et al 2022, Zhang et al 2024, and Kazlou et al 2023). We note that the literature is not monolithic regarding the recognition of CCS growth and injectivity limits as a key constraining criterion for carbon storage in deep mitigation futures. In particular, Ringrose and Meckel (2019) conclude that:

"Using this analysis, it is clear that the required well rate for realizing global CCS in the 2020–2050 timeframe is a manageable fraction of the historical well rate deployed from historic petroleum exploitation activities and is most efficiently achieved with multiple simultaneous regional developments."

We agree with the reviewers that it is important to elevate and highlight a discussion of injectivity limitations and feasibility. At the same time, we believe that our manuscript adds to the literature in a useful and constructive way in addition to the literature on feasibility considerations of injection growth and rate maxima, as recognized by a number of the reviewers.

We propose and have implemented the following considerations of injection rates which we believe appropriately address the reviewers very useful comments:

1. We highlight explicitly in the Methods that part of our calculation which translate storage volumes to CO₂ stored includes consideration of pressure-based

injectivity limits (referencing original sources of Kearns et al 2017 and Szulczewski et al 2012), which we felt was not clear to reviewers and would not have been clear to readers.

2. We include the following explicit reference to more explicit injectivity considerations in our discussion of limitations of our assessment, citing the appropriate literature:

While we focus on volumetric limits, a large body of recent literature also highlights concerns regarding the technoeconomic feasibility of scaling up subsurface injection to the levels shown in future scenarios (17, 22, 60, 84, 85), with most scenarios breaching assessed feasibility limits, though some argue that these levels are geophysically feasible (86). More explicit consideration of injection feasibility would likely further reduce our estimate of usable storage potential.

3. We include a new SI Figure 13 with indicative thresholds resulting from the literature for relevant limits due only to scale-up rates to raise to the reader both important volumetric and scale-up/injectivity rate concerns

Response related to the planetary boundary framing

We very much appreciate that this issue was raised and have reflected critically on it. After careful consideration, we agree that it is fundamentally a weakness of our original draft to frame the paper in terms of a planetary boundary. As is rightly pointed out, the body of literature and its critiques around the planetary boundary framework could lead to potentially unfounded critiques of our paper and can be avoided by reframing the discussion. We have adjusted our presentation to refer to "planetary limits" and removed the discussion and comparisons with planetary boundaries. We feel that it is important to highlight that what we have developed here is an analytical framework and analysis to holistically assess geological, technical, and risk-based limits to carbon storage and hope that this is an acceptable compromise in the framing of this approach.

Responses to Individual Reviewer Comments

To organize this section, we present the original comments we received in blue text. We then interweave direct responses as necessary which are not covered by our above thematic responses in black.

Referee #1 (Remarks to the Author):

This paper extends the concept of planetary boundaries to the amount of geological storage available. In principle, this could be a valuable contribution to the literature if

framed in a way that focuses on the risk of needing to rely so heavily on carbon dioxide removal as a climate stabilization method rather than the potential shortcomings of geological storage capacity. However, as written, I have three major concerns with the paper.

We strived to include in the original manuscript a clear introduction, framing, and discussion around carbon storage as a depletable resource which presents risks when relying too much on CDR or maintaining residual emissions. To further elevate this discussion, we have added statements related to competition with CDR in key locations, including:

- " Planning to use storage for both reducing sources of emissions and for CDR presents risks should storage infrastructure fail to be deployed at scale. This risk can be somewhat mitigated by sustainably deploying a portfolio of approaches, including storing carbon at scale in the built environment through what is referred to as the forestry-construction pump (*15*), storing carbon in minerals through enhanced rock weathering, enhancing the carbon content of soils, and conserving and expanding land and coastal carbon sinks, which would reduce dependence on geologic storage by using less-durable carbon storage media. . "

- Demand for geologic carbon storage can also increase in these scenarios based on other mitigation choices, such as deploying blue (fossil-gas based) vs. green (renewable-energy based) hydrogen, electrifying steel production vs. capturing carbon from existing processes, and reducing demand for cement vs. capturing carbon from cement production. Even more storage will be needed than shown in scenarios if the earth system responds differently than expected by the current state-of-the-art climate models (*16*).

- We argue that nations should make explicit plans for geologic carbon storage for both mitigating continued sources of fossil-fuel emissions as well as durably removing and storing CO₂. Treating geologic carbon storage as a limited global resource for use which should be managed on an intergenerational time scale requires considering tradeoffs of continuing to emit carbon from fossil-based sources versus utilizing storage space for removing carbon from the atmosphere to ultimately lower global mean temperature for this and future generations.

- For example, understanding whether nations plan to maximize their use of storage resources for abating continued sources of emissions that could be avoided (e.g., through the pursuit of blue, fossil-based hydrogen and fossil CCS) or strategically minimize the dependence of their climate strategies on carbon storage (e.g., by deploying green, renewable-based hydrogen, other renewable energy strategies, and minimal CDR) will enhance understanding of the robustness of mitigation plans.

First, there is **little that is novel** in the paper that has not been discussed in previous literature, including much of the work going back to the IPCC special report on Carbon Capture and Storage, and extensive work by authoritative organizations such as the U.S. Geological Survey. It has long been recognized that not all the geological storage "resource" would be available due to economic, environmental, and social concerns. The methodology of using exclusion layers is also not novel, and in fact, a standard approach for assessing capacity in basins.

We respectfully disagree with the reviewer, noting that reviewers 2 & 4 both see the novelty of our approach and analysis specifically of moving past purely technical potential of geological carbon storage capacity.

We agree with the reviewer that there is a significant strand of existing literature (which we cited in the original manuscript and now cite even more broadly). However, this literature is rarely consistent in how the total volume of storage capacity is defined. For example, in the study of Wei et al. (2021), a global map of the distribution of sedimentary basins is used coupled with the capacity calculations of the US Department of Energy to give an estimate of storage capacity. The sedimentary basins used in this study are not limited in type, location or depth to basement. In the study by Zhang et al. (2024) they cite a few references for past studies of storage potential, however these are mostly grey literature. A key paper cited is the study by Kearns et al. (2017) that has similarities with our methodology for estimating storage potential. The study of Kearns does not however limit basins by storage depth nor question zones where storage may not be possible due to environmental, social and economic concerns. Our goal here is not to claim novelty in how we define our carbon storage limits, but to apply them in an open and documented manner to stress how they will limit CSS plans.

We observe that this statement is specifically related to lines 85-87 in the original manuscript. We have adjusted those lines as follows to make clearer the novelty of our analysis, approach, and framing:

" While previous studies have estimated global (11, 27) or regional (28, 29) technical storage potential, to date, no consistent estimate of global carbon storage potential assesses this variety of different risk factors to determine the available storage potential from a precautionary harm-prevention perspective as expected under the UNFCCC (30). . "

To place our study in the context of past efforts to estimate carbon storage potential, we have conducted a literature review of other assessments, which we summarize in the attached table (literature_review.xlsx).

Second, some of the "boundaries" are overly stringent. For example, excluding oil and gas resources on the basis that they are used for EOR does not make sense. The ratio of CO₂ to oil recovery can be adjusted so that more CO₂ is stored per barrel of oil than would be released to the atmosphere when it is burned. Furthermore, in a world of

declining demand for oil, depleted fields may provide a low cost and secure option for CO₂ storage. In this case, the biggest (and significant) risk is that old wells leak CO₂ to shallower depths, and even possibly to the surface. Nevertheless, excluding depleted oil and gas reservoirs is not scientifically justified on the basis that is currently used with EOR. The cut-off depth of 2,500 m depth is also arbitrary. There are many places where the basement rock is far greater than a depth 2,500 m and therefore, risks of seismic activity associated with basement rocks not a significant concern. Similarly, constraining water depth to 300 m is also arbitrary. Off-shore rigs are available today that can drill in much deeper water than this cut-off. If some or all these limitations are relaxed, CO₂ storage capacity could easily double, bringing into question the conclusion that “this carbon storage potential limit sets a cap of about 0.7°C on the total warming that can ever be sustainably reversed if the full prudent potential is used for CO₂ removal.”

We thank the reviewer for their comment, but we do not state anywhere in our manuscript that we exclude oil and gas resources or basins. In fact, we explicitly include them and point to them being the most likely areas where carbon storage will take place (see, e.g., Figure 3).

Regarding concerns about our (a) assumptions around ocean depth and cut-off depth and (b) sensitivity around our estimate of storage and temperature drawdown, we refer the reviewer to our above discussion.

My third concern stems from the framing of the paper which suggests that the capacity will be far less than is needed, rather than addressing the more important point that relying too much on carbon dioxide removal for climate stabilization is a risky strategy. With the “planetary boundary capacity” of 1,500 Gt, that would be sufficient to capture and store 15 Gt per year (300 times more than the 50 Mt per year of geological storage today) for 100 years. If the “planetary boundary” capacity is double this, which is likely based on the above discussion, the “cooling potential” would double to 1.4 oC, enabling an enormous 30 Gt/year (equivalent to 83% of today’s fossil fuel emissions) for 100 years. Accomplishing this degree of scaleup and paying the associated costs for carbon dioxide removal will be an enormous challenge in and of itself. In light of this, the risk of not accessing this storage capacity due to financial limitations and lack of political will, are a far greater than the risk of not having the storage capacity we need. As written, this paper is most likely to have the effect of further discouraging use of the capacity we do have, particularly in the short run, for reducing emissions from industrial sources such as cement, steel, ammonia, where few other options exist today.

We have striven in the original manuscript to refrain from overt policy prescriptiveness. That is to say, we in no way claim that there is far less capacity than is needed. Rather, we highlight that our analysis implies that countries need to be explicit about how much carbon storage they plan to use and that, if countries were to follow global mitigation

pathways as assessed by the IPCC, that they would likely, in aggregate, surpass the limit we assess in the 2100s. We note that what level of carbon storage is needed is highly dependent on mitigation strategies pursued and assumptions around costs, technology availability, etc. A core and critical outcome of our paper, as discussed thoroughly therein, is that policy makers and energy system planners must recognize and address the critical tradeoffs between mitigation strategies which will need to utilize carbon storage resources.

Additionally, please see our earlier response where we try to judicially balance arguments for using the available storage space between reducing emissions from sources vs. carbon removal.

We do not disagree with the reviewer that there is risk that lack of political will results in under-utilization of geologic storage resources. However, policy makers are at the moment setting near and long-term goals which implicitly rely on large-scale geologic carbon storage. We argue these assumptions and their associated tradeoffs are critical and should be made explicit. Overall, while we appreciate the reviewer's concern regarding the framing of our paper, we think it is important for policy makers to understand that these tradeoffs exist and that they are made explicit in future climate targets and mitigation strategies.

Referee #2 (Remarks to the Author):

General comments:

Using geospatial analysis combined with various risk exclusion layers the authors estimate a "prudent" boundary for CO₂ storage in sedimentary basins. They estimate that the prudent potential is around 1,500 GtCO₂, a value that could well be exceeded this century in emission scenarios that rely heavily on CO₂ storage to compensate for residual emissions and to achieve net negative CO₂ emissions.

This is the first study that presents an estimate of geological storage that goes beyond technical potential. It is an important contribution as it changes the narrative from CO₂ storage being an unlimited resource. Rather, it shows that prudent CO₂ storage is limited, implying that **it needs to be used strategically to satisfy various needs.** The study is topical as remaining emission scenarios that are consistent with limiting warming to the Paris Agreement climate goal require carbon dioxide removal and storage, with geologic storage being the only permanent solution.

We thank the reviewer for their comment recognizing the novel addition to the literature our study provides.

The methodology is straightforward and clearly explained. My main reservation is the lack of uncertainty analysis – the authors present a best estimate of the prudent storage potential without an uncertainty range. It is hard to believe that there is no uncertainty in the

estimation of storage potential in sedimentary basins and the data used in the various risk exclusion layers.

We agree with the reviewer and refer them to our thematic response to uncertainty quantification and analysis assumptions at the top of this response.

The study is logically structured and well written but could benefit from further elaboration and clear definitions in some instances. Figures could be used to more clearly illustrate the points described in the text – figure references are not always obvious.

Specific comments:

I. 21: “We establish a prudent planetary boundary of around 1500 Gt of CO₂ storage”. What is the uncertainty for this estimate?

I. 24: “Can at most reduce global temperature by 0.4-0.7°C”. This is a highly optimistic estimate. As the authors mention in the manuscript, the temperature reduction is likely smaller due to inertia effects and asymmetries in the climate-carbon cycle response.

We agree that this estimate represents an upper limit, and have edited the statement in that regard. It now says: “. Fully using geologic storage for carbon removal caps possible global temperature reduction to 0.4-0.7°C (0.35-1.2°C). ”

However, given the limitations in the current evidence base, we are unable to add a strong implication on the climate-carbon response asymmetry, and, in absence of new evidence, have decided to diligently describe the uncertainties based on the latest assessment of the IPCC AR6 Working Group 1, which considers all identified studies. See also our response to a related comment below.

I. 41: Define CCS.

Done

I. 56-59: Clarify that CDR with biological storage is not permanent (i.e. storage timescale is less than several hundred to a thousand years).

Done

I. 177: “physical storage potential of 11,784 GtCO₂”. What is the uncertainty around this estimate?

We are unfortunately limited by our primary data sources (e.g., Evenicks et al and Kearns et al) which do not report uncertainty estimates. We have attempted to address uncertainty where possible (see above comment).

I. 181-187: “Many countries...”: The points described in the text cannot easily be seen in Fig. 2. I suggest to add a panel that shows the total (onshore+offshore) prudent potential for countries.

We have now added country estimates in our supplementary information (Table S5).

I. 187-189: If I read Fig. 2c correctly, it suggests that the largest restriction is from maximum storage depth, not Arctic storage.

This was a mistake in our original manuscript and has been updated to read: “When we apply exclusion layers in the order presented here, we find that that the largest increase in storage would be realised if our assumptions regarding storage and ocean depth were relaxed, followed by assumptions regarding storage in polar regions and protected areas (Table S1).

I. 194-197: Suggest to rewrite in less technical language.

We have tried to simplify the language in this paragraph. The goal of this section is to explain at a high level the main reasons why future mitigation scenarios and strategies deploy carbon capture, which we hope the below edit helps convey in a more straightforward manner:

" The majority of mitigation strategies consider geologic carbon storage to some extent in support of the transformation towards net-zero and net-negative CO₂ futures (51). Which carbon capture approaches are utilized in future mitigation scenarios depends on a variety of factors, including assumed costs, scale up rates, and the efficiency of capture. Coupling carbon capture with hydrogen and synthetic fuel production provides efficient pathways to achieve deep mitigation in heavy industry and transportation sectors (52) and can enable net-negative sectoral outcomes when using biomass instead of fossil feedstocks (53). Future scenarios tend to utilize large-scale carbon capture at individual point sources in the power (e.g., biomass and fossil-fueled generation) and industry (e.g. cement production) sectors due to the relatively high concentrations of carbon in the effluent flue gases (54). Net-negative emissions futures are increasingly being studied that utilize Direct Air Capture with CCS, which removes and durably stores ambient CO₂ from the atmosphere (55–58). The regional allocation of the ultimate storage depends on assumptions of regional storage capacity and infrastructure needs, which vary in their level of detail across different modelling frameworks (59, 60)."

I. 217: Unclear how the bullets relate to Fig. 3a.

We have adjusted the reference to Figure 3a earlier in the sentence to better reflect the portion of the sentence it refers to

I. 231 (and elsewhere): Clarify what percentages (33%, 50%) refer to.

This percentage is in relation to temperature exceedance following the approach used by the IPCC in its recent AR5, AR6, and Special Reports. We have clarified here and elsewhere what this percentage refers to.

I. 269, 277: The calculation assumes that the TCRE is the same for CO₂ emissions and removals, which is not the case due inertia effects and asymmetries in the climate-carbon cycle

response to CO2 emissions and removals. The authors allude to this asymmetry later in the manuscript (l. 343-347), but the implications of this asymmetry should already be discussed here. An additional relevant reference is Zickfeld et al., 2016.

We fully acknowledge the possibility of inertia effects (in particular, the Zero Emissions Commitment, ZEC) and asymmetries in the climate-carbon cycle response to CO2 emissions and removals, and contextualize our findings with reference to the latest assessments on this topic (Lee et al. 2021, Palazzo-Corner et al. 2023) which report a consensus of near-zero additional warming after zero CO2 emissions, albeit with an uncertainty range that goes in both directions. Individual other studies can fall in different parts of this uncertainty range, but are often by themselves not conclusive. For example, the modelling results of Zickfeld et al. (2016) show that their modelling setup exhibits a positive ZEC, and the inertia effects illustrated in that study can therefore be understood in that context. Equally, another seminal study on the asymmetry between emissions and removals (Zickfeld et al. 2021) provides important insights for the Earth system response from an equilibrium state, and identifies asymmetry in carbon-climate response that is state-dependent (with simulations starting from about 2-times pre-industrial CO2 concentrations showing an asymmetry where removals are more effective in cooling than emissions in warming in their modelling framework). Given these uncertainties, our preferred way forward is to acknowledge the remaining uncertainties while contextualizing these with the assessment of the latest IPCC AR6 Working Group 1 report, which superseded, and therefore included the insights of both of the above-mentioned asymmetry studies.

I. 323: Is “preventative” the same as “prudent”? Suggest to use consistent terminology.

The term "preventative" is directly linking to the text of the UNFCCC and the Paris Agreement. However, we have tried to streamline our use of the terms prudent and preventative, keeping "preventative" in the context of international agreements and otherwise trying to maintain our use of "prudent". We have adjusted this sentence accordingly.

I. 343: See additional reference provided in earlier comment.

Thank you for pointing to this study. We have considered it and refer to all evidence on this topic as assessed in the latest report of the IPCC AR6 Working Group 1 contribution.

Fig. 2: i) Some country ISO codes are not obvious. Suggest to use country names in radial diagrams. ii) Clarify what unit the technical storage potential is given in in the radial diagrams (numbers in parentheses) iii) consider including a panel that shows total (onshore + offshore) storage potential), which is referenced in the text. To make room the radial diagrams could be removed – Fig. 2d illustrates countries’ storage loss in a more intuitive way.

We have updated Fig 2 such that this no longer applies.

Fig. 3: Clarify what is meant by “preventative limit” in the legends. Is it the same as the prudent planetary boundary? Clarify what percentages (33%, 50%) mean.

We have updated Fig 3 labels.

Fig. 4: Units are missing from horizontal axes.

We have updated Fig 4 units.

References:

Zickfeld, K., A.H. MacDougall and H.D. Matthews, 2016, On the proportionality between global temperature change and cumulative CO₂ emissions during periods of net negative CO₂ emissions, *Environmental Research Letters*, 11, 055006.

<https://iopscience.iop.org/article/10.1088/1748-9326/11/5/055006>

Referee #3 (Remarks to the Author):

The manuscript by Gidden et al. present a global CO₂ storage capacity assessment based on different geological, social, environmental and regulatory/legal aspects that produce a prudent (as in not-so-optimistic) storage potential. The authors use this new assessment to constrain the capacity of CO₂ storage in achieving climate targets. **The paper is well written and, to the best of my knowledge, the implementation of the capacity assessment within the IPCC scenarios is not an issue.** My review focuses mainly in the aspects regarding the analysis of CO₂ storage potential, as it is closer to my field of expertise.

We thank the reviewer for their extensive and useful review. We appreciate that the reviewer found the manuscript well written and in particular our scenario analysis to be of high quality. We will focus on the remaining critiques in this response.

I have a number of concerns regarding the authors' assessment of CO₂ storage potential. Please find also attached the manuscript with further comments and questions.

- My main concern is that the authors **have not included the effect of the injectivity in their assessment**, probably the most important factor beyond volumetric capacity in terms of accessibility to storage resources. There is a high uncertainty in the estimation of injectivity values, because it is necessary to have a good knowledge of the subsurface (something that can be difficult/expensive, especially in hydrocarbon-poor provinces), but this information can really impact the calculations reducing the overall optimistic volumetric capacity calculations. The risk of reduced injectivity can probably be more damaging to the development of CCS than other factors mentioned by the authors (e.g. the risk of induced seismicity); **low injectivity-related issues have impacted CO₂ storage projects before, and have even led to shutting down entire projects (see the discussion in Huerta et al., 2020)**. The lack of robust injectivity information was already flagged in the work by Lane et al. (2021), which point out that “deep uncertainty over the sustainable injection rate at any given location will constrain the pace and scale of carbon capture and storage deployment”. **Grant et al (2021) further discuss this issue, and urge for the inclusion of injection rates in the capacity estimations – and the subsequent IAMs to which they input – and I strongly agree with this view. Ringrose and**

Meckel (2019) already attempted to produce a global assessment considering injectivity, where they derived generalised pressure-depth functions to produce a proxy for injectivity. Lane et al (2021) noted that these type of injectivity analyses can also be misrepresentative, because injectivity may decay with time, but you could incorporate a similar assessment to approach this issue. **IMHO, the injectivity aspect is crucial to actually produce a more reliable high-level capacity estimate** and create the impact that the authors are looking for. **Without it, I do not see that extra relevance of the analysis made**, especially considering the **arbitrariness** in some of the criteria used in the capacity assessment.

Given that both Reviewers 1 and 3 raise issues around injectivity we have chosen to respond collectively to these critiques. In particular, we agree with the reviewers that injection capacity is a critical issue to consider when assessing future mitigation strategies, that it has been recently covered in the literature (which we cite), and that we provide a complementary perspective advancing the state of knowledge regarding volumetric potentials. We refer Reviewer 3 to that section at the top of our response.

- The authors claim that “to date, no consistent estimate of global carbon storage potential has included these various factors” (line 85 in the manuscript), but there are plenty of such capacity assessments in the literature, perhaps not applying the exact same criteria, but with similar approaches. **See for example the dedicated IEA analysis (Malischek and McCulloch, 2021), the study by Wei et al. (2021) or the references in Zhang et al (2024).** In that sense, the authors should discuss what are the main differences between their methodology and the ones already published, and to analyse the differences between the storage capacities calculated using their assessment and the other examples in literature. I think making this comparison it would be useful to put your assessment into perspective.

We respectfully disagree with the reviewer, noting that reviewers 2 & 4 both see the novelty of our approach and analysis specifically of moving past purely technical potential of geological carbon storage capacity.

We agree with the reviewer that there is a significant strand of existing literature (which we cited in the original manuscript and now cite even more broadly). However, this literature is rarely consistent in how the total volume of storage capacity is defined. For example, in the study of Wei et al. (2021), a global map of the distribution of sedimentary basins is used coupled with the capacity calculations of the US Department of Energy to give an estimate of storage capacity. The sedimentary basins used in this study are not limited in type, location or depth to basement. In the study by Zhang et al. (2024) they cite a few references for past studies of storage potential, however these are mostly grey literature. A key paper cited is the study by Kearns et al. (2017) that has similarities with our methodology for estimating storage potential. The study of Kearns does not however limit basins by storage depth nor question zones where storage may not be possible due to environmental, social and economic concerns. Our goal here is not to claim novelty in how we define our carbon storage limits, but to apply them in an open and documented manner to stress how they will limit CSS plans.

We observe that this statement is specifically related to lines 85-87 in the original manuscript. We have adjusted those lines as follows to make clearer the novelty of our analysis, approach, and framing:

" While previous studies have estimated global (11, 27) or regional (28, 29) technical storage potential, to date, no consistent estimate of global carbon storage potential assesses this variety of different risk factors to determine the available storage potential from a precautionary harm-prevention perspective as expected under the UNFCCC (30). ."

To place our study in the context of past efforts to estimate carbon storage potential, we have conducted a literature review of other assessments, which we summarize in the attached table (literature_review.xlsx).

- In general, the authors mix absolute criteria (e.g. inside or outside protected areas, 0-1) with more interpretative criteria. The water depth is a clear example of this mixing, as it is currently set at 300m – 1500m depth, but **the authors fail to offer a convincing reason to why a platform located at 301m or 1501m should be disregarded**. Water depth in itself is not necessarily an excluding factor (e.g. in opposition to the lack of a competent seal), it may "just" make the whole project more expensive (it may make it too expensive or risky, but that is a case-specific issue). There are other factors that can make a project expensive/riskier (and also too expensive/risky) that are not included in the analysis, for example **the depth of the reservoir** (the deepest cut off it is currently only considered from a safety point of view) or the distance from the CO2 sources (e.g. longer routes may lead to more incidents, plus increased transportation costs). **I suggest that the modelling should focus on absolute (as in binary criteria, qualifying or disqualifying) factors, and leave out other more interpretative (or case specific) factors**. If the aim is to provide absolute thresholds/range of storage capacity potential (as in a screening exercise), then perhaps it would be more appropriate to use the entire range in relative criteria and afterwards use the sensitivity to include the authors' criteria (i.e. to give some weight to the analysis).

As the reviewer helpfully suggests, we have now split our analysis between binary factors and those that lie on a continual spectrum. We have extensively updated our analysis to include multiple dimensions of uncertainty across the various non-binary factors we consider. We are limited based on our GIS assessment approach to still assess discrete values but back our selection of parameters with updated and robust observational evidence or literature reviews where appropriate. We refer Reviewer 3 to that section at the top of our response.

- Storage in transboundary basins seems to me another arbitrary factor to be considered. Indeed, there can be issues associated with transboundary storage (same as with other types of activities), but there are also opportunities for collaboration and even cost reduction (and hence reduced risk) in transboundary zones, a clear example being the North Sea, where transport and storage shared across countries is the key to the development of the Northern Lights project (<https://www.equinor.com/news/20220829-important-step-to-decarbonise-europe>). Having issues associated to the operation in transboundary regions is thus not a direct disqualifying criterion, in my opinion. Also, there are other important factors that are not

considered in the analysis; for example, **if CCS is banned in certain country** is definitely a more restraining aspect, and potentially with a stronger effect in the capacity assessment than the transboundary issues. Another aspect not considered within this criterion is the interaction between the zones of influence of the storage (e.g. the pressure plume), which can sometimes extend tens of kilometres beyond the plume of CO₂; it is more likely that storage operation can perturb the pressure state in a contiguous reservoir than the plume of CO₂ trespassing the boundary. **The 22.2 km is the distance from the shore to the international waters, but this is not necessarily a limitation for economic activities in the vicinity of these boundaries (e.g. fisheries or wind farms) so why would they be a limitation for CCS?**

We should clarify that we do not include a 22.2km boundary offshore in our main analysis. Instead, we assume that nations have full access to sedimentary basins within their offshore Exclusive Economic Zones, consistent with existing treaties such as the London Protocol. We include a 22.2km buffer zone between onshore geopolitical boundaries only as a sensitivity to our primary analysis to help the reader understand what the effect on total storage potential would be should countries not collaborate in transboundary storage. We include a smaller buffer in our primary analysis given the current state of geopolitical agreements, and also include a no-buffer sensitivity (see Table S2). In particular, we have expanded our discussion of this assumption in our methods section as follow:

“The choice for 6 nautical miles is derived from the definition of territorial waters boundary description within UNCLOS, which sets the territorial waters to be within 12 nautical miles. This assumption would effectively mean that sedimentary basin resources along the international boundary up to 11.1 km on both sides will not be used for injection and storage of CO₂, irrespective of the depth of storage and angle of approach to the storage site. We recognize that the pressure plume associated with CO₂ storage would reach further than our assumed limit of 6 nautical miles which is a limitation to our analysis should this consideration be built into future transboundary storage agreements. In addition to the central case, we include sensitivities of this exclusion layer to cover full 22.2 km (12 nautical miles) and no buffer considerations on both sides of the national boundaries (Fig S6).”

With regards to the review’s point regarding the potential limiting effect of CCS bans on carbon storage potential, we conducted a review of the bans/major restrictions currently in force in various countries around the world (see Table S4). This review showed that (a) not many countries currently have bans or major restrictions in force, and (b) many of them are in the process of removing or reducing these restrictions – although these developments remain politically contested. We have added a sentence and supporting references in the revised manuscript highlighting this :

“For example, CCS is currently banned or majorly restricted in some European countries (see Table S4 for current countries with restrictive CCS policies), but there are growing discussions to adjust the existing regulations to allow onshore and offshore storage in order to achieve climate targets. However, these policy developments remain politically

contested, highlighting the volatile and uncertain nature of public and political support of geologic carbon storage. “

Other minor issues (some are also highlighted in my comments within the manuscript):

- This is perhaps a language issue, but **the authors use “carbon” and “CO2” interchangeably** when talking about geological storage. I am not a native English speaker, but I would be careful in the usage of these terms; I understand that “carbon” is broadly used in the climate/environmental field, but the only type of carbon that is stored in CCS is carbon dioxide. Because of this lack of consistency in the terms used, it is sometimes difficult to discern to which one are the authors referring throughout the text. For example, when referring to the planetary boundary concept introduced in line 106, does it refer to carbon in general (as in the C cycle) or to CO₂ in particular? The authors should be consistent throughout the manuscript to avoid misinterpretations.

We have adjusted language in the text to refer to carbon when discussing long-term (millennial-scale) storage including mineral trapping (including when referring to the first C in Carbon Capture and Storage) and to use CO₂ when directly referring to carbon dioxide removal, atmospheric release of CO₂, and units relevant for climate assessment and policy (including storage units).

- The authors report global capacity estimations of 10,000-40,000 Gt CO₂ from literature, with industry estimates of around 14,000 Gt CO₂, but their starting capacity is on the lower end of this range (11.784 Gt CO₂). **The calculation of this initial geological potential should be better described in the Methods.** Does the sedimentary basin data analysed already include the depths? Also, the authors should delve onto this discrepancy, which clearly impact the final assessment.

We have expanded our methods discussion to better establish our initial potential assessment.

References:

Grant, N., Gambhir, A., Mittal, S., Greig, C., & Köberle, A. C. (2022). Enhancing the realism of decarbonisation scenarios with practicable regional constraints on CO₂ storage capacity. *International Journal of Greenhouse Gas Control*, 120, 103766.

Huerta, N. J., Cantrell, K. J., White, S. K., & Brown, C. F. (2020). Hydraulic fracturing to enhance injectivity and storage capacity of CO₂ storage reservoirs: Benefits and risks. *International Journal of Greenhouse Gas Control*, 100, 103105.

Lane, J., Greig, C., & Garnett, A. (2021). Uncertain storage prospects create a conundrum for carbon capture and storage ambitions. *Nature Climate Change*, 11(11), 925-936.

Malischek, R. & McCulloch, S. - IEA (2021), The world has vast capacity to store CO₂: Net zero means we'll need it, IEA, Paris <https://www.iea.org/commentaries/the-world-has-vast-capacity-to-store-co2-net-zero-means-we-ll-need-it>

Ringrose, P. S., & Meckel, T. A. (2019). Maturing global CO₂ storage resources on offshore continental margins to achieve 2DS emissions reductions. *Scientific reports*, 9(1), 1-10.

Wei, Y. M., Kang, J. N., Liu, L. C., Li, Q., Wang, P. T., Hou, J. J., ... & Yu, B. (2021). A proposed global layout of carbon capture and storage in line with a 2 C climate target. *Nature Climate Change*, 11(2), 112-118.

Zhang, Y., Jackson, C., & Krevor, S. (2024). The feasibility of reaching gigatonne scale CO₂ storage by mid-century. *Nature Communications*, 15(1), 6913.

[please also find pdf with comments attached]

We thank the reviewer for a detailed annotated pdf. We have reviewed it thoroughly and tried to address all comments in our resubmission.

Referee #4 (Remarks to the Author):

This paper seeks to establish a framework for a planetary boundary for geological CO₂ storage. It does so by performing a spatially explicit analysis that excludes areas based on depth, active seismic zones, protected and polar areas, areas of human settlement, and transboundary basins. **The paper will be of wide interest. It is novel, and uses data appropriately, employing a useful methodological approach.** The only issues I would identify relate to the presentation, clarity, and context.

We thank the reviewer for their comment recognizing the novel addition to the literature our study provides.

There is one issue that should be addressed, regarding the clarity of the introduction, and a second issue that I would invite the authors to reflect on, regarding the suitability of the planetary boundaries framework.

1. The introduction should be reworked for clarity around the basic concepts and activities being described.

Both the abstract and the first few paragraphs have some conflation between negative emissions / CDR and geologic storage / CCS. For example, the abstract begins with three sentences about geologic storage; then the fourth sentence talks about both near-term emissions reductions (including CCS, or not?) as well as “durable carbon dioxide removal”, which should probably be “carbon dioxide removal that uses geological CO₂ storage” for accuracy.

We have adjusted language in the text to refer to carbon when discussing long-term (millennial-scale) storage including mineral trapping and to use CO₂ when directly referring to carbon dioxide removal and units relevant for climate assessment and policy (including storage units).

It will be critical to be very clear about the distinction up front, and throughout the paper. It is brought up somewhat in the third paragraph, but still a bit confusing for readers (and the third paragraph actually works better as the second paragraph — because the first paragraph is about expectations, and so is the third — better not to tack back and forth). The fix is as simple as having a sentence along the lines of “Geologic storage is necessary for many CDR approaches, such as DACCS and BECCS” — but also just having clear basic definitions of CDR,

CCS, and storage. It will also be important to define “durable” CDR as being not ocean alkalinity enhancement or enhanced weathering (I guess – although if a direct ocean capture company is able to successfully make solid carbonates, wouldn’t that also be durable CDR, albeit without CO₂ injection into sedimentary rock?)

This is a very important comment, and we have striven to better identify and separate CCS/CDR and net-zero/net-negative emissions and relevant mitigation strategies.

In particular, we have added transition text in the introduction to highlight:

"Net-zero CO₂ emissions will occur when gross sources of CO₂ equal removal by sinks. Carbon capture and storage (CCS) plays a role in both reducing sources (through, e.g., storing captured CO₂ from cement production and fossil-fuel combustion) and durably removing CO₂ from the atmosphere (e.g., storing CO₂ captured from the atmosphere or biomass combustion). "

Have clarified our point regarding durability and further clarified net-zero and net-negative outcomes “This risk can be somewhat mitigated by sustainably deploying a portfolio of approaches, including storing carbon at scale in the built environment through what is referred to as the forestry-construction pump (15), storing carbon in minerals through enhanced rock weathering, enhancing the carbon content of soils, and conserving and expanding land and coastal carbon sinks, which would reduce dependence on geologic storage by using less-durable carbon storage media.”

And have further clarified in our section on mitigation strategies:

“The majority of mitigation strategies consider geologic carbon storage to some extent in support of the transformation towards net-zero and net-negative CO₂ futures (51). Which carbon capture approaches are utilized in future mitigation scenarios depends on a variety of factors, including assumed costs, scale up rates, and the efficiency of capture. Coupling carbon capture with hydrogen and synthetic fuel production provides efficient pathways to achieve deep mitigation in heavy industry and transportation sectors (52) and can enable net-negative sectoral outcomes when using biomass instead of fossil feedstocks (53). Future scenarios tend to utilize large-scale carbon capture at individual point sources in the power (e.g., biomass and fossil-fueled generation) and industry (e.g. cement production) sectors due to the relatively high concentrations of carbon in the effluent flue gases (54). Net-negative emissions futures are increasingly being studied that utilize Direct Air Capture with CCS, which removes and durably stores ambient CO₂ from the atmosphere (55–58). The regional allocation of the ultimate storage depends on assumptions of regional storage capacity and infrastructure needs, which vary in their level of detail across different modelling frameworks (59, 60). “

In lines 61 – 68, it is again rather unclear if the authors are talking about CCS or geologic storage. The first sentence reads as if CCS for point-source / mitigation, the rest of the

paragraph seems to talk about geologic storage for CDR. And the reader doesn't really have a sense of what percentage of storage is for point-source capture vs. CDR in the scenarios being discussed. **It would be really nice to add a figure illustrating that context for people (i.e., in most scenarios, is more storage from point-source mitigation or from CDR, and how does that change over time)? Do the takeaways here point towards not just avoiding excessive CDR, but to avoiding fossil gas or industrial decarb strategies with CCS?** The latter is not really present in the paper. Given that we are unlikely to even do much CDR, the takeaways of this analysis for industrial decarbonization in sectors like steel or hydrogen could be better foregrounded. **You could imagine a version of the paper that was centered on the need to better incentivize green hydrogen vs. blue hydrogen**, for example.

In particular for lines 61-68, we have clarified as:

"The scale of deployment of CCS (and thus geologic storage) in future scenarios is not absolute and depends on policy and political choices."

We have also added a new SI figure (Fig S7) and sentence related to how scenarios use storage volume:

"Scenarios tend to use storage primarily for carbon removal and fossil point-source capture, with industrial capture playing an important but smaller role (Fig S7)."

While we see the point regarding future CDR as an opinion of the reviewer, we agree that it is important to discuss the use of geologic storage in relation to different mitigation strategies as the reviewer nicely points out. We have adjusted the text in a number of ways to take this comment on board:

- Introduction (within lines 61-68): "Demand for geologic carbon storage rises in these scenarios based on other mitigation strategies, such as deploying blue (natural-gas based) vs. green (renewable-energy based) hydrogen, electrifying steel production vs. capturing carbon from existing processes, and reducing demand for cement vs. capturing carbon from cement production. "

- Conclusion: " For example, understanding whether nations plan to maximize use storage resources by abating continued sources of emissions (e.g. through blue hydrogen and fossil CCS) or strategically minimizing dependence on carbon storage (e.g., by deploying green hydrogen, other renewable energy strategies, and minimal CDR) will enhance understanding of the robustness of mitigation plans."

The authors might also want to clarify what the literature says about the potentials of non-geologic-storage CDR approaches. These findings could be read in two ways that the authors do not mention. One, the findings could be read as implying the need for CDR that doesn't involve geologic storage, such as ocean alkalinity enhancement or enhanced weathering or ocean fertilization. Two, there is a reading that 1.5°C is even more unachievable without solar radiation modification. You may even end up with people resurrecting that idea of just injecting CO₂ into the deep ocean, if you define all these onshore areas as too risky. **I am not suggesting tackling all that in the paper, but it would be helpful for readers to acknowledge that there are more methods of CDR that don't hinge on geologic storage.**

In line 58, it is mentioned that the “forestry-construction pump” would reduce dependence on geologic storage — why mention that one (I’m skeptical about this being of climate-significant scale), and not the others that people are currently enthusiastic about?

We have added additional CDR methods that are currently actively being discussed in both the academic and grey literature. We have adjusted this section as follows:

" Many scenarios that limit climate change to goals set out by governments in the Paris Agreement (14) assume a large scale up of the use of CCS, which contributes both to abating further combustion of fossil fuels, reducing emissions from industrial sectors that have limited or no mitigation alternatives, as well as providing storage for CO₂ that was removed from the atmosphere contributing to Carbon Dioxide Removal (CDR). Planning to use storage for both reducing sources of emissions and for CDR presents risks should storage infrastructure fail to be deployed at scale. This risk can be somewhat mitigated by sustainably deploying a portfolio of approaches, including storing carbon at scale in the built environment through what is referred to as the forestry-construction pump (15), storing carbon in minerals through enhanced rock weathering, enhancing the carbon content of soils, and conserving and expanding land and coastal carbon sinks, which would reduce dependence on geologic storage by using less-durable carbon storage media. "

2. Is “planetary boundaries” the right framework for this paper? What is gained by this, vs. framing it just as a right-sizing estimates of storage using a holistic risk assessment, or “a more realistic view of storage capacity given social and geologic factors”, or a frame around “equitable division of storage reserves and the need for planning”, etc?

There are two main issues with employing this framework.

Issue 1 - As I read it, the “framework establishes global boundaries, which, if transgressed, substantially increase the risk of a change in Earth system functioning and/or its long-term state and trajectory” (from Rockstrom et al. 2024 review article, ‘Planetary Boundaries guide humanity’s future on Earth’, in Nature Reviews Earth & Environment). So this seems like the framework is focused on the boundaries that risk the change in Earth system functioning, i.e. breaching the boundaries poses a global risk. But it seems like with geologic storage, doing more of it and breaching the boundary poses a bunch of local risks? I believe this has been a longstanding critique of the planetary boundaries framework – “its inclusion of subglobal-scale, nonsystemic, and aggregate processes that have shown little evidence of threshold behavior so far”, as Biermann and Kim (2020) identified in their ARES review article on the “boundaries of the planetary boundary framework”. I know that the authors related to planetary boundaries have also stated that no global thresholds are assumed within the framework, but it still seems like a point of confusion, and I wouldn’t expect all the readers of the paper to be familiar with the nuances and debates around the framework.

Issue 2 - It seems that another one of the main critiques of the planetary boundaries framework is that the setting of boundaries is arbitrary, and more of a political exercise or social construction rather than something derived from the scientific literature on thresholds. This relates to what Biermann and Kim (2020) identify as the critique of boundaries derived from Science and Technology Studies – that “boundaries are proposed by experts without stakeholder consultation or public participation and deliberation.” I mean, the parameters in the study are the authors expert opinion, not developed through deliberation.

In short, the “planetary boundaries” idea suggests a biophysical limit, but the parameters here include social judgements of risk. For example, in lines 82 – 98 – this paragraph moves from talking about perceptions of harm and risk to talking about planetary limits, and then in line 106-107, the definition is explicitly about harm prevention and risk avoidance. So this reads as basically a social construction and not a biophysical limit. Doesn't this kind of undo the discursive power of the planetary boundaries framework (that it relies on science)?

Whether or not a person wants to live within 25 km of a storage well can be a matter of social circumstance, but it can also be a social judgement based on one's values. I have met people in places like Texas who live literally next door to CO₂ injection wells that they can see from their yard and were not troubled by this. Other people may have no practical choice about where to live due to income and other factors, but still, what that buffer should be is a question for society. **While public opinion of CO₂ storage is low today, future people may prefer CO₂ storage to alternatives of warming or solar geoengineering,** if those end up being the choices on offer. Similarly, whether or not it is anticipated that nation-states can agree on managing a basin is a governance matter rather than a biophysical one. The choice about the Arctic circle is also a social choice; e.g. there may be actors in Alaska who are scoping storage above the Arctic circle and see that as desirable and workable. We already know Santos is interested in CO₂ storage on the North Slope with the Pikka project, and ConocoPhillips is also scoping North Slope sequestration. **So the value choices of these authors might not be the same as companies or even Alaskan voters, potentially.** Same thing for seismic hazards — there are already areas in S2 fig (a) (note typos on this figure) that are being explored or developed for storage. It seems that regulators think they can manage this through things like the Class VI framework in the US. **The authors may differ in their opinion, but society may choose something else.**

I think that critics of the planetary boundaries framework would read this paper and say, see, here's one more piece of evidence that planetary boundaries are just a made up expert construction and not grounded in biophysical reality. Do the authors want to arm the planetary boundary critics in that way?

We thought our response to this series of comments was sufficiently important to merit raising to the top for all reviewers to be aware of. We repeat our response here for ease for Reviewer 4.

We very much appreciate that this issue was raised, and we have reflected critically on it. After careful consideration, we agree that it is fundamentally a weakness of our original draft to frame the paper in terms of a planetary boundary. As is rightly pointed out, the body of literature and its critiques around

the planetary boundary framework could lead to potentially unfounded critiques of our paper and can be avoided by reframing the discussion. We have adjusted our presentation to refer to "planetary limits" and removed the discussion and comparisons with planetary boundaries. We feel like it is important to highlight that what we have developed here is an analytical framework and analysis to assess geological, technical, and risk-based limits to carbon storage and hope that this is an acceptable compromise in approach.

Personally, I would find the paper more convincing without this framing. **It is already interesting in terms of providing a more realistic assessment of CO2 storage potential than is indicated in the literature, and highlighting the tradeoffs between using storage for CCS vs. CDR.** To me, mixing it up with the planetary boundaries framework just confuses the issue. People are already critical of scientists for using the language and institutions of science to defend their value judgements and personal preferences. I would give any framing that might appear that way a wide berth in these times, when public trust in scientific neutrality is paramount.

We thank the reviewer again for their thoughtful and deliberative comments. We believe our paper has been significantly improved because of these discussions.

Final Response to Reviewers for: A prudent planetary limit for geologic carbon storage

Overview

Reviewers provided useful feedback for this final revision round. We extract below specific text need response or comment and have provided our responses in blue.

Specific Responses

Referee #2 (Remarks to the Author):

The authors clearly put considerable effort into revising the manuscript. However, one of my main comments was not addressed to my full satisfaction. The authors considered uncertainties regarding the criteria used in the exclusions layers in their analysis but missed to report and discuss the resulting range in the prudent planetary limit (1.285 to 2,710 GtCO₂; Table S2) in the main manuscript. Given that relaxing constraints (within uncertainty bounds) on some of the exclusion criteria nearly doubles the prudent storage limit, acknowledging these uncertainties in the context of storage requirements in mitigation scenarios appears warranted.

I. 21, I.191, I. 286, Fig. 2b: Report full uncertainty range for the prudent planetary storage limit.

We have now included the range from Table S1 in the abstract (first paragraph) and main text. We have further updated Figure 2(b) to include additional sensitivity ranges.

I.288, “based on assessed uncertainties”: clarify that you are referring to uncertainties in exclusion criteria.

We have added this.

I. 290-291, I. 299: In precise terms this quantity is referred to as the “transient climate response” to cumulative CO₂ emissions.

We agree that this is the precise term, but also feel that there is usefulness for general readability (of temperature response to cumulative emissions).

I. 292; “similar” should read “similar in magnitude”.

We have made this change.

Table S2: Include in extended data file for ease of access.

Per instructions, we will include this as an external data file with a DOI.

Referee #3 (Remarks to the Author):

Reviewer 3 provided a useful discussion, but did not provide any actionable items that we could take forward. We appreciate Reviewer 3's engagement with our manuscript.

Referee #4 (Remarks to the Author):

Reviewer 4 also provided mostly a narrative response. We thank Reviewer 4 for their engagement with our paper.

The revision also addresses some of my concerns about the clarity of the distinctions being made between CCS and CDR. Very minor point, I'm not sure that storing carbon in minerals through enhanced rock weathering would be "less durable" (line 65), though it would be harder to measure. On a writing level, I am glad to see the argument mentioned up front in the introduction.

Here, we agree with the reviewer that ERW is a more durable option (not currently deployed at any scale and being discussed only academically). The purpose of the sentence is to highlight the difference between scalability of current options and durability of those options, so we think it best to just remove the reference to ERW and keep the rest of the sentence.